# *In vivo* antagonistic role of the Human T-Cell Leukemia Virus Type 1 regulatory proteins Tax and HBZ

**Abdou Akkouche**[1,2], **Sara Moodad**[1,2], **Rita Hleihel**[1,2], **Hala Skayneh**[1,2],
**Séverine Chambeyron**[3], **Hiba El Hajj**[4]*, **Ali Bazarbachi**[1,2]*

**1** Department of Internal Medicine, Faculty of Medicine, American University of Beirut, Beirut, Lebanon,
**2** Department of Anatomy, Cell Biology and Physiological Sciences, American University of Beirut, Beirut,
Lebanon, **3** Institute of Human Genetics, CNRS, UMR 9002, Montpellier University, Montpellier, France,
**4** Department of Experimental Pathology, Immunology and Microbiology, Faculty of Medicine, American
University of Beirut, Beirut, Lebanon

osThese authors contributed equally to this work.
* he21@aub.edu.lb (HEH); bazarbac@aub.edu.lb (AB)

org/10.1371/journal.ppat.1009219

London, UNITED KINGDOM

**Data Availability Statement:** All relevant data
(Figures) and Supporting Information files are
within the submitted manuscript.

## Abstract

Adult T cell leukemia (ATL) is an aggressive malignancy secondary to chronic infection by
the human T-cell leukemia virus type 1 (HTLV-1) infection. Two viral proteins, Tax and HBZ,
play central roles in ATL leukemogenesis. Tax expression transforms T cells *in vitro* and
induces ATL-like disease in mice. Tax also induces a rough eye phenotype and increases
hemocyte count in *Drosophila melanogaster*, indicative of transformation. Among multiple
functions, Tax modulates the expression of the enhancer of zeste homolog 2 (EZH2), a
methyltransferase of the Polycomb Repressive Complex 2 (PRC2), leading to H3K27me3-
dependent reprogramming of around half of cellular genes. HBZ is a negative regulator of
Tax-mediated viral transcription. HBZ effects on epigenetic signatures are underexplored.
Here, we established an *hbz* transgenic fly model, and demonstrated that, unlike Tax, which
induces NF-κB activation and enhanced PRC2 activity creating an activation loop, HBZ nei-
ther induces transformation nor NF-κB activation *in vivo*. However, overexpression of Tax or
HBZ increases the PRC2 activity and both proteins directly interact with PRC2 complex
core components. Importantly, overexpression of HBZ in *tax* transgenic flies prevents Tax-
induced NF-κB or PRC2 activation and totally rescues Tax-induced transformation and
senescence. Our results establish the *in vivo* antagonistic effect of HBZ on Tax-induced
transformation and cellular effects. This study helps understanding long-term HTLV-1 per-
sistence and cellular transformation and opens perspectives for new therapeutic strategies
targeting the epigenetic machinery in ATL.

## Author summary

Adult T cell leukemia-lymphoma is an aggressive hematological malignancy, caused by
the retroviral infection with HTLV-1. Tax and HBZ play critical roles in leukemia devel-
opment. Tax activates the NF-κB pathway and modulates the epigenetic machinery to

**Funding:** This work was supported by the American University of Beirut Medical Practice Plan (AB and HEH), the University Research Board (AB and HEH), the Lebanese National Council for Scientific Research (AB and HEH), the Lady TATA Memorial Trust (AA), unrestricted grants from Novartis (AB), Roche (AB) and Takeda (HEH), The Fondation pour la Recherche Médicale (DEQ20180339167) (SC). The funders had no role in study design, data collection and analysis, decision to publish, or preparation of the manuscript.

**Competing interests:** The authors have declared that no competing interests exists.

induce cellular proliferation and malignant transformation. We generated *hbz* or *tax/hbz* transgenic fly models and explored the phenotypes and epigenetic changes *in vivo*. Unlike Tax, HBZ expression failed to activate NF-κB or to induce transformation or senescence *in vivo*, yet activated PRC2 core components resulting in subsequent epigenetic changes. HBZ expression in *tax* Tg flies inhibits Tax-induced NF-κB or PRC2 activation, resulting in inhibition of malignant cellular proliferation and its consequent senescence. Our study proves the antagonistic effect of HBZ on Tax-induced transformation *in vivo*, providing further understanding on ATL pathogenesis.

## Introduction

Adult T-cell Leukemia/Lymphoma (ATL) is an aggressive hematological malignancy secondary to chronic infection with the human T-cell leukemia virus type 1 (HTLV-1) [1,2]. The viral oncoprotein Tax alters many cellular pathways [3,4], transforms T cells *in vitro*, and induces leukemia in transgenic mice [5–9]. Tax also induces transformation in a transgenic *Drosophila melanogaster* model [10]. The role of Tax in initiating cellular transformation is well established, yet its role in maintaining the leukemic phenotype is more controversial. This is mostly due to the undetectable Tax protein levels in most circulating HTLV-1 infected or ATL leukemic cells [11–13], likely due, at least in part, to its strong immunogenic properties eventually leading to rapid elimination of Tax expressing cells by the host immune system [14,15]. Furthermore, persistent Tax-induced NF-κB activation results in cellular senescence [16,17]. Nevertheless, ATL cells share the same phenotype of Tax expressing cells [18]. Recent studies demonstrated that transient bursts of Tax expression occur sequentially in small fractions of HTLV-1 infected or ATL-derived cells [19]. Although Tax protein expression is very low in most patients with ATL, long-term survival of the bulk of ATL cells may depend on transient bursts of Tax expression in some, if not the majority, of ATL cells and/or in HTLV-1 infected non-malignant cells [20,21].

A small percentage of HTLV-1 infected individuals develops ATL after a long latency period of several decades following infection, suggesting that Tax-facilitated accumulation of subsequent genetic changes might play a major role in final transformation [22–24]. As a matter of fact, multiple somatic mutations were reported in ATL cells [25]. In addition to these genetic events, a key role for epigenetic changes was recently unraveled. Indeed, early after viral infection, Tax activates the transcription of key components of the polycomb-repressive complex 2 (PRC2). These mainly include the enhancer of zeste homolog 2 (EZH2), in addition to suppressor of zeste 12 homolog (SUZ12), and the embryonic ectoderm development (EED) [26,27]. Transcriptional activation is mediated, at least in part, by NF-κB [28]. Dysregulation of PRC2 was reported in various types of cancers due to its ability to affect the expression of genes involved in cell survival, proliferation, or apoptosis [29]. In ATL, Tax interaction with PRC2 results in global alteration of the tri-methylation of the histone 3 on the lysine 27 (H3K27me3), a repressive histone mark, leading to epigenetic reprogramming of more than half of cellular genes, including that of important genes for leukemogenesis [28].

Another viral protein, HBZ, is encoded by the complementary strand of HTLV-1 [30]. Unlike Tax, HBZ is expressed in all HTLV-1 carriers and ATL patients [31]. *In vitro*, HBZ promotes cellular proliferation [32] and affects several cellular pathways [33], including c-jun, junB, and junD [34,35]. Interestingly, biological functions were described for both HBZ protein and mRNA. While HBZ transcripts enhance the proliferation of T cells, HBZ protein counterpoises Tax-induced effects. For instance, HBZ decreases Tax expression by inhibiting

its transcription [36]. HBZ also inhibits Tax-induced activation of the classical NF-κB pathway. In that sense, HBZ inhibits the DNA binding potential of RelA, while also promoting the degradation of this activating subunit of the NF-κB pathway [32]. Through this decrease in the activity of the classical NF-κB pathway, HBZ counteracts Tax-induced persistent and constitutive NF-κB activation and its resulting cellular senescence response [16,17]. Furthermore, HBZ is a negative regulator of Tax-mediated viral transcription [30]. Yet, knock-down of HBZ only results in a modest inhibition of ATL cells proliferation [37,38].

The fruitfly, *Drosophila melanogaster* emerged as a potent *in vivo* model for studying cancer, and particularly the implication of epigenetic signature in cancer. In that sense, polycomb group proteins were originally discovered in *Drosophila* where their mutations resulted in improper body plans [39]. Moreover, the PRC2 complex core components are conserved between Human (EZH1/2, SUZ12, EED, RBBP4/7) and *Drosophila* (E(z), Suz(12), Esc/EScI, Nurf55/Caf1) [40], validating this model to study the epigenetic mediated implications of this complex in cancer, including leukemia.

In this study, we established an *hbz* transgenic fly model and assessed the role of HBZ on epigenetic modifications. Contrary to *tax* expression which both induces NF-κB activation and enhances PRC2 activity creating an activation loop, HBZ did not result in cellular transformation *in vivo*. Nevertheless, transgenic flies expressing HBZ exhibited increased PRC2 activity and H3K27me3, similar to that seen in *tax* transgenic flies. Interestingly, overexpressing HBZ in *tax* transgenic flies rescued Tax-induced cellular transformation and prevented Tax-mediated NF-κB activation and subsequent senescence. This antagonistic role of HBZ on Tax effects was confirmed in human ATL-derived cells, in HEK293T or CD4 T cells transiently expressing both viral genes. Our results provide an *in vivo* evidence of the ability of HBZ to interact with PRC2 components and to attenuate Tax-induced transformation, at least in part, through epigenetic modulation. Yet, in the absence of Tax, HBZ maintains PRC2 activation. Our study clearly demonstrates the *in vivo* effect of HBZ on Tax-induced transformation and allows further understanding on the role of Tax and HBZ in HTLV-1 persistence and transformation and highlights the potential advantage of epigenetic therapy in ATL.

## Results

### HBZ does not induce transformation in a fruit fly model

To establish an *in vivo hbz* transgenic model, we used the same approach as in our previously reported *tax* transgenic fly model [10]. The expression of *hbz* was induced, using the UAS-Gal4 system under the GMR-Gal4 promoter, which drives HBZ expression in all differentiated photoreceptor cells posterior to the morphogenetic furrow [41]. We then compared the transformation potential of HBZ and Tax in transgenic flies, by examining the ommatidial structures in the eyes. The eye phenotypes were evaluated and scored, through evaluating the severity of loss of ommatidia, in addition to the mechano-sensory bristle alignment, misplacement, or duplication [10,42]. Consistent with our previous findings [10], *tax* transgenic flies exhibited a rough eye phenotype indicative of cellular transformation (Fig 1A and 1B), whereas only 32% of *hbz* transgenic flies (12 out of 37 eyes) displayed minimal changes in the normal ommatidial structure compared to the control flies, while the remaining ones did not show any phenotypic differences when compared to the control flies (Fig 1A and 1B). Tax and HBZ expression in these respective transgenic flies was confirmed by quantitative real-time RT-PCR and Western blot (Figs 1C and S1A), and HBZ functional activity was confirmed by the higher JunD transcript levels in *hbz* transgenic flies, as compared to *tax* transgenic flies or control flies (Fig 1D). We also examined the impact of HBZ on *Drosophila* hematopoietic system. Tax or HBZ expression in hemocytes were induced under the control of the hemocyte

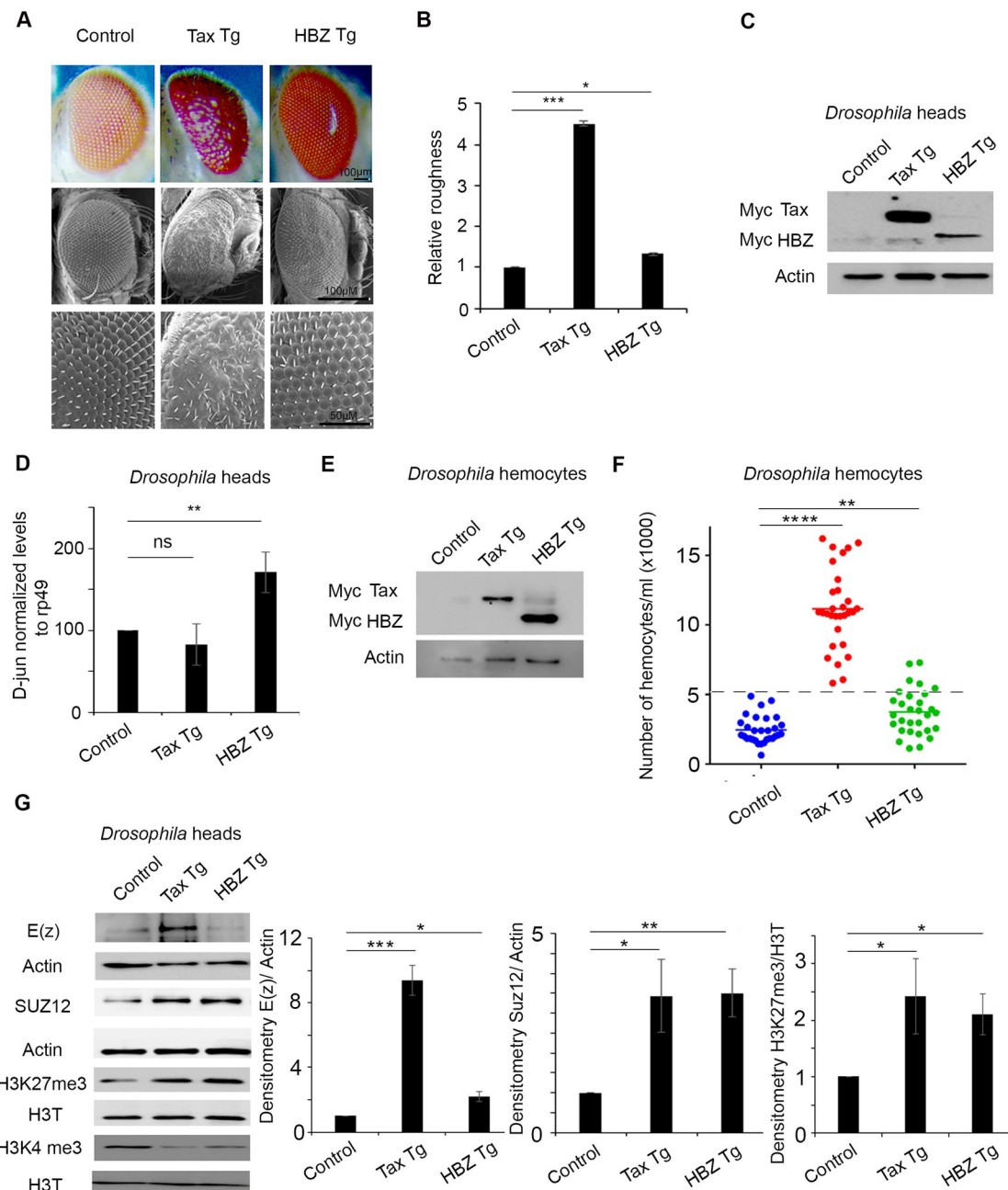

**Fig 1. Overexpression of HBZ *in vivo* activates the PRC2 complex but does not induce transformation.** Representative light microscopy images (Scale bar 100μm) and scanning electron microscopy images (Scale bar 100μm and 50μm) of adult eyes from transgenic flies expressing Tax (GMR-Gal4>UAS-Tax) (Tax Tg), or HBZ (GMR-Gal4>UAS-HBZ) (HBZ Tg), under the control of the eye-specific GMR promoter (GMR-GAL4). GMR-GAL4>*w1118* was used as control. **(B)** Relative roughness was quantified based on the number of ommatidial fusions and the extent of bristle organization (n = 37, from three independent crosses). $p<0.001$ (***) and $p<0.05$ (*). **(C)** Cell lysates (150 μg) from control, Tax Tg and HBZ-Tg transgenic adult flies heads were analyzed by western blotting and probed with anti-Myc or anti-actin antibodies. Genotypes indicated are under the control of eye specific promoter GMR-GAL4. **(D)** Levels of expression of D-jun in the transgenic adult flies heads as indicated. Transcript levels were normalized to Rp49. Reported values are the average of three independent experiments and error bars represent SD of triplicates. (ns = not significant), $p<0.01$ (**). **(E)** Cell lysates (300 μg) from control (HMLΔ-Gal4>*w1118*), Tax Tg (HMLΔ-Gal4>UAS-Tax) and HBZ-Tg (HMLΔ-Gal4>UAS-HBZ) transgenic larvae were analyzed by western blotting confirming the expression of Tax and HBZ transgene in larval hemocytes. Genotypes indicated are under the control of the hemocyte-specific promoter (HMLΔ-GAL4). **(F)** Circulating haemocytes were counted in control larvae (HMLΔ-Gal4>*w1118*), larvae expressing Tax (HMLΔ-Gal4>UAS-Tax) (Tax Tg) or HBZ (HMLΔ-Gal4>UAS-HBZ) (HBZ Tg) (n = 30, from three independent crosses). Data represent mean ± standard error of the mean. $p<0.01$ (**), $p< 0.001$ (***). **(G)** Cell lysates (150 μg) from control

(GMR-GAL4>*w1118)*, Tax Tg (GMR-Gal4>UAS-Tax), or HBZ Tg (GMR-Gal4>UAS-HBZ) transgenic adult flies heads were analyzed by western blotting and probed with indicated antibodies. Densitometry histograms represent an average of 3 independent experiments.

specific promoter HMLΔ-Gal4, using the UAS-Gal4 system, and confirmed by quantitative real-time RT-PCR and Western Blot (Figs 1E and S1B). Consistent with our previously reported data [10], third instar larvae of *tax* transgenic flies exhibited a significant increase in the number of circulating hemocytes compared to control flies (Fig 1F). On the other hand, HBZ-expression led to a minor increase (5 out of 30 larvae) in the number of circulating hemocytes (Fig 1F). Yet, this minor increase was much below that obtained in larvae of *tax* transgenic flies. These results are consistent with the obtained phenotypes in the eyes of respective transgenic flies, and represent a conclusive *in vivo* evidence, that HBZ, contrary to Tax, has a weak oncogenic potential *in vivo*.

## Both Tax and HBZ activate key components of PRC2 complex leading to H3K27me3 accumulation

While Tax is known to alter PRC2 activity, the putative link between HBZ and PRC2 remains underexplored. We took advantage of the fruit fly system to characterize the epigenetic modulation by Tax and HBZ *in vivo*. In line with previously reported data in human lymphocytes [28], Tax expression in *Drosophila* correlated with increased levels of both E(z) and SUZ12 proteins, two components of the PRC2 complex (Fig 1G). Concomitant with this increase, the repressive mark H3K27me3 was also increased. This was consistent with the decreased level of the activation mark H3K4me3 (Fig 1G). Unlike in *tax* transgenic flies, E(z) protein levels were not affected in *hbz* transgenics (Fig 1G), while SUZ12 protein levels were higher, as compared to control flies (Fig 1G). Increased SUZ12 protein levels following the overexpression of HBZ, were similar to observed in *tax* transgenics and also correlated with an increase in H3K27me3 and a decrease in H3K4me3 (Fig 1G). Altogether, these findings demonstrate that Tax and HBZ enhance the accumulation of the repressive H3k27me3 histone mark through modulating different components of the PRC2 complex.

## Tax-induced transformation requires NF-κB activation and is partially dependent on PRC2 activity

Excessive PRC2 activity and inappropriate deposition of H3K27me3 repressive mark, are critical determinants of the abnormal transcriptome of various cancers [43,44]. Despite the increased activity of PRC2 in both *tax* and *hbz* transgenic flies, transformation was only observed in *tax* flies. This prompted us to dissect the role of PRC2 activity in Tax-induced transformation, *via* knocking-down E(z) and SUZ12 in *tax* transgenics. Tax expression in both knockdown flies was confirmed by Western blot (S2A Fig). E(z) and SUZ12 respective knockdowns were confirmed by quantitative RT-PCR (S2B Fig). Interestingly, downregulation of E(z) and SUZ12 rescued, only partially, Tax-induced rough eye phenotype (Fig 2A and 2B) and Tax-induced increase in hemocyte count (Fig 2C). Since Tax is a potent activator of the NF-κB pathway [45], we analysed the expression of *Relish*, a transcription factor in the *Drosophila* NF-κB pathway [46], and one of its well-established downstream transcriptional targets, *Diptericin* [47]. Consistent with our previous data [10], the transcript levels of both Relish and Diptericin were significantly increased in *tax* transgenic flies (Fig 2D and 2E). Similarly, Relish protein levels were also increased (Fig 2F). To confirm the role of NF-κB activation in the transformation phenotype, we knocked-down Relish in Tax-expressing flies. As previously

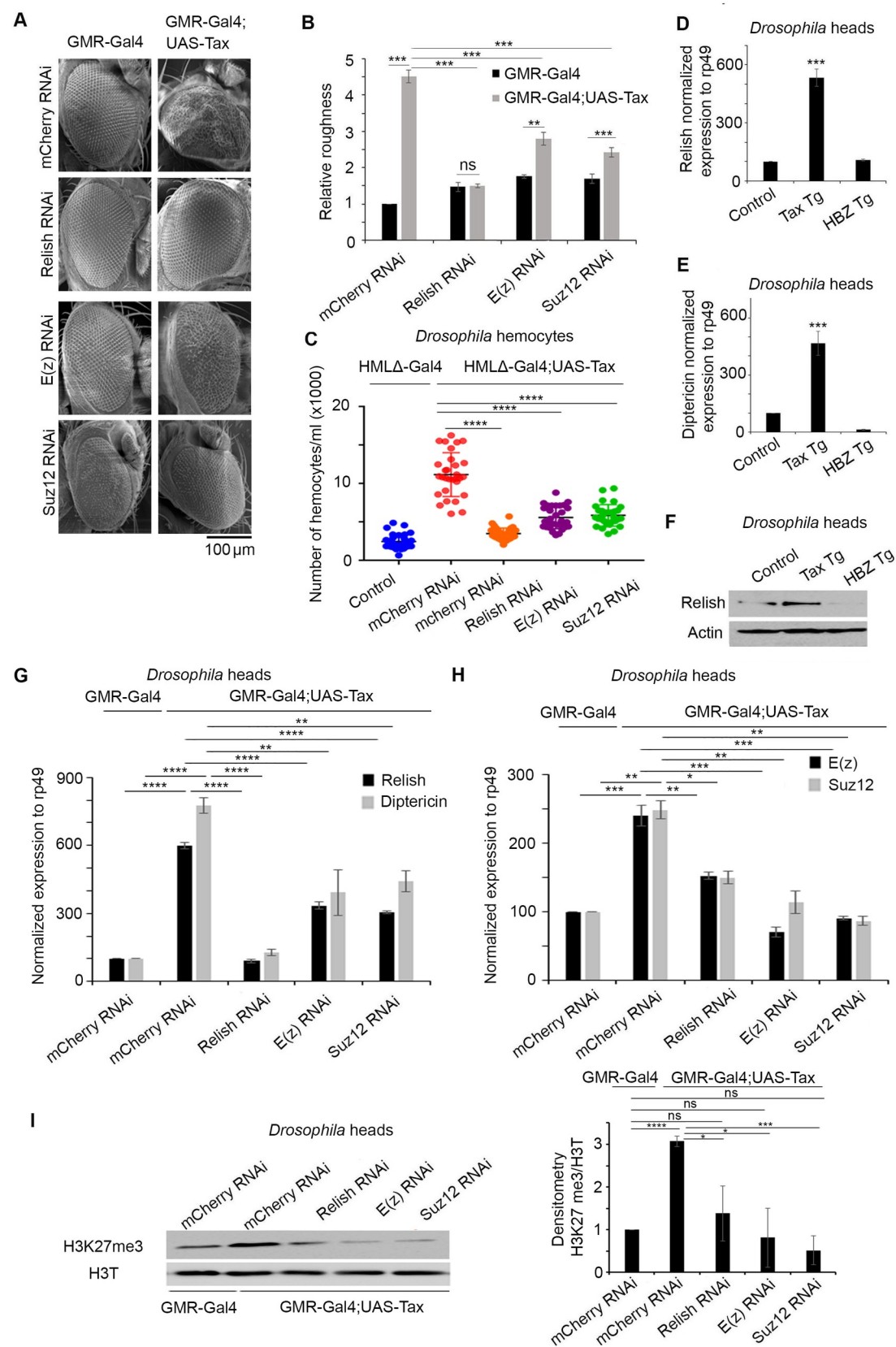

**Fig 2. Tax-induced transformation *in vivo* is NF-κB and PRC2 dependent. (A)** Relish, E(z), and SUZ12 expression was inhibited by RNAi in flies overexpressing Tax in the compound eye. Representative scanning electron microscopy images of adult eyes are shown. Scale bar 100μm. **(B)** Relative roughness was quantified based on the number of ommatidial fusions and the extent of bristle organization (n = 30, from three independent crosses). (ns = not significant), p<0.01 (**), p<0.001 (***). **(C)** Relish, E(z), and SUZ12 expression was inhibited by RNAi in flies overexpressing Tax in the hemocytes. Circulating hemocytes were counted in control larvae (HMLΔ-Gal4> mCherry RNAi), (HMLΔ-Gal4; UAS-Tax>mCherry RNAi), (HMLΔ-Gal4;UAS Tax>Relish RNAi), (HMLΔ-Gal4;UAS-Tax>E(z) RNAi) and (HMLΔ-Gal4;UAS-Tax>Suz12 RNAi) (n = 30, from three independent crosses). Data represent mean ± standard error of the mean. p<0.0001 (****). **(D)** Levels of expression of Relish in the control (GMR-GAL4>*w1118)*, Tax Tg (GMR-Gal4>UAS-Tax), or HBZ Tg (GMR-Gal4>UAS-HBZ) transgenic adult flies heads. The values were normalized to *Rp49*. Reported values are the average of three independent experiments and error bars represent SD of triplicates. p<0.001 (***). **(E)** Levels of expression of Diptericin, a Relish target gene, in the control (GMR-GAL4>*w1118)*, Tax Tg (GMR-Gal4>UAS-Tax), or HBZ Tg (GMR-Gal4>UAS-HBZ) transgenic adult flies heads. The values were normalized to Rp49. Reported values are the average of three independent experiments and error bars represent SD of triplicates p<0.001 (***). **(F)** Cell lysates (150 μg) from control (GMR-GAL4>*w1118)*, Tax Tg (GMR-Gal4>UAS-Tax), or HBZ Tg (GMR-Gal4>UAS-HBZ), transgenic adult flies heads were analyzed by western blotting and probed with anti-Relish or anti-actin antibody. **(G)** Transcript expression levels of Relish and Diptericin in the control (GMR-Gal4>mCherry RNAi), Tax control (GMR-Gal4;UAS-Tax>mCherry RNAi), Tax/Relish RNAi (GMR-Gal4;UAS-Tax>Relish RNAi), Tax/E(z) RNAi (GMR-Gal4;UAS-Tax>E(z) RNAi) and Tax/Suz12 RNAi (GMR-Gal4;UAS-Tax>Suz12 RNAi) transgenic flies as indicated. Transcript levels were normalized to Rp49. Reported values are the average of three independent experiments and error bars represent SD of triplicates. p<0.01 (**), p< 0.0001 (****). **(H)** Transcript levels of expression of E(z) and Suz12 in in the control (GMR-Gal4>mCherry RNAi), Tax control (GMR-Gal4;UAS-Tax>mCherry RNAi), Tax/Relish RNAi (GMR-Gal4; UAS-Tax>Relish RNAi), Tax/E(z) RNAi (GMR-Gal4;UAS-Tax>E(z) RNAi) and Tax/Suz12 RNAi (GMR-Gal4;UAS-Tax>Suz12 RNAi) transgenic flies as indicated. Transcript levels were normalized to Rp49. Reported values are the average of three independent experiments and error bars represent SD of triplicates. p<0.01 (**), p< 0.001 (***). **(I)** Cell lysates (150 μg) from the control (GMR-Gal4>mCherry RNAi), Tax control (GMR-Gal4;UAS-Tax>mCherry RNAi), Tax/Relish RNAi (GMR-Gal4; UAS-Tax>Relish RNAi), Tax/E(z) RNAi (GMR-Gal4;UAS-Tax>E(z) RNAi) and Tax/Suz12 RNAi (GMR-Gal4;UAS-Tax>Suz12 RNAi) transgenic adult flies heads were analyzed by western blotting and probed with H3K27me3 antibody. Densitometry histograms represent an average of 3 independent experiments. (ns = not significant), p<0.05 (*), p<0.001 (***), p<0.0001 (****).

reported [10], Relish silencing completely rescued the rough eye phenotype (Fig 2A and 2B) and completely reverted Tax-induced increase in hemocyte count (Fig 2C), suggesting that NF-κB activation is mandatory for Tax-induced transformation *in vivo*. Consistent with the lack of HBZ transformative potential *in vivo*, Relish and Diptericin exhibited no changes in *hbz* transgenic flies, when compared to control flies (Fig 2D–2F), indicating that HBZ does not activate the NF-κB pathway *in vivo*.

We then explored a potential link, if any, between Tax-mediated NF-κB activation and PRC2 activation. For that purpose, we used the following transgenic fly lines: Tax/Relish RNAi (defective for NF-κB pathway), Tax/E(z) RNAi and Tax/SUZ12 RNAi (defective for PRC2 pathway), and Tax/mcherry RNAi as a control. Importantly, knock down of Relish totally reverted NF-κB activation, and significantly decreased E(z) and SUZ12 transcript levels (Fig 2H). Conversely, knock down of E(z) or SUZ12 only partially reverted Tax-induced NF-κB activation (Fig 2G). Concurrent with these observed differences of transcription in either knockdown lines, a significant decrease in H3k27me3 level was observed in Tax/Relish RNAi, Tax/E(z) RNAi and Tax/SUZ12 RNAi flies confirming a link between NF-κB and PRC2 complex (Fig 2I).

Altogether, these results implicate PRC2 core components E(z) and SUZ12 in Tax-mediated cell proliferation, and NF-κB activation and demonstrate that Tax-induced NF-κB activation enhanced PRC2 activity, hence creating an activation loop between both pathways.

## HBZ expression in *tax* transgenic flies abrogates NF-kB and PRC2 activation and reverses Tax-induced transformation

To explore the direct effect of HBZ co-expression on Tax-induced cellular changes *in vivo*, we generated transgenic flies that overexpress both HBZ and Tax. Tax and HBZ expression were confirmed in protein extracts from *Drosophila* heads by Western blot (Fig 3A). Unlike *tax*

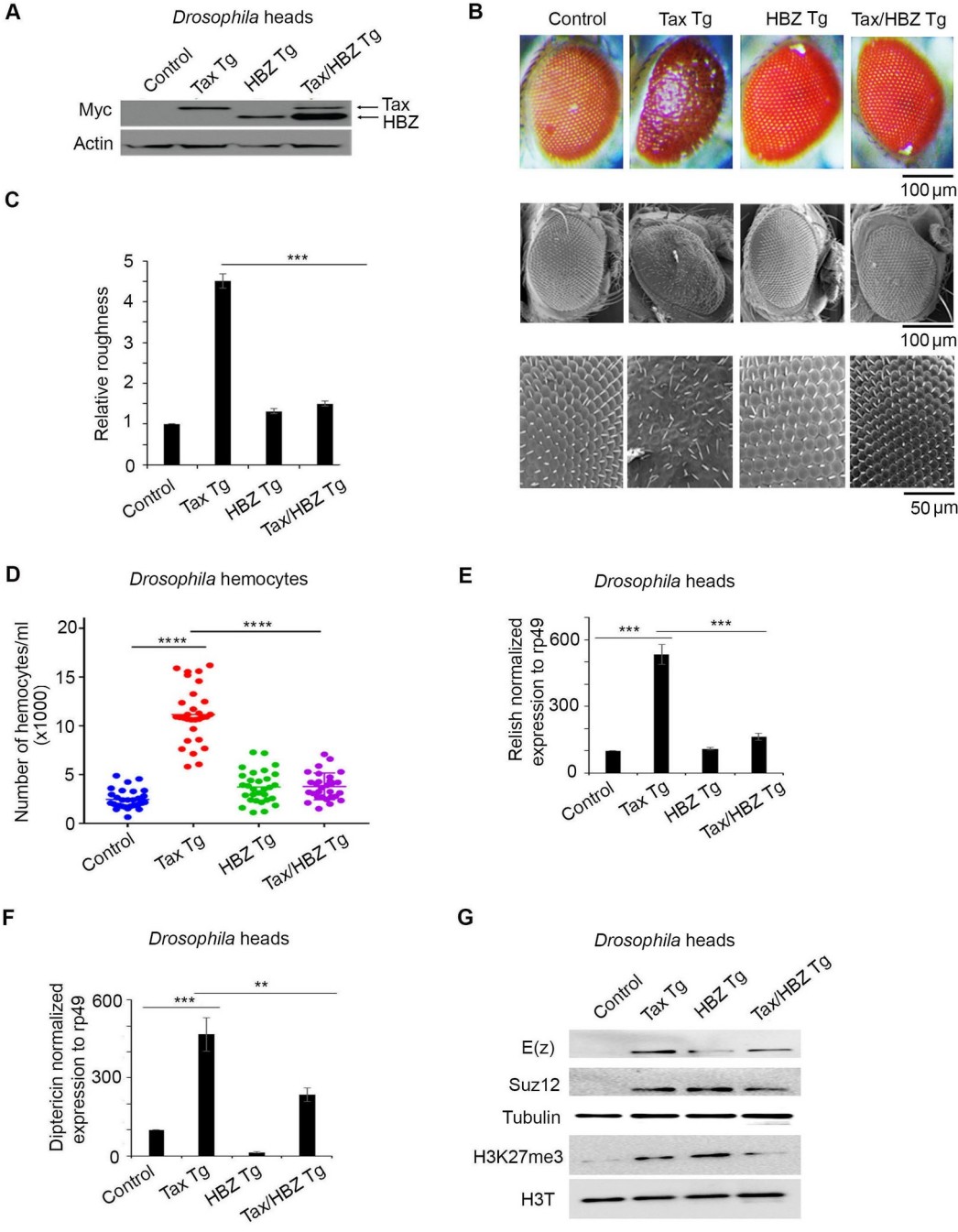

**Fig 3. HBZ overexpression abrogates Tax-induced NF-kB, PRC2 activation and *in vivo* transformation. (A)** Cell lysates (150 μg) from control (GMR-GAL4>*w1118)*, Tax Tg (GMR-Gal4>UAS Tax), HBZ Tg (GMR-Gal4>UAS-HBZ) or Tax/HBZ Tg (GMR-Gal4; UAS-Tax>UAS-HBZ), transgenic adult flies heads were analyzed by western blot confirming the expression of *tax* and *hbz* transgenes. **(B)** Representative light microscopy images (Scale bar 100μm) and scanning electron microscopy images (Scale bar 100μm and 50μm) of adult eyes are shown. **(C)** Relative roughness was quantified based on the number of ommatidial fusions and the extent of bristle organization (n = 30, from three independent crosses). p<0.001 (***). **(D)** Circulating hemocytes counted, in control larvae (HMLΔ-Gal4>*w1118*), larvae expressing transgenic Tax alone (HMLΔ-Gal4>UAS-Tax), HBZ alone (HMLΔ-Gal4>UAS-HBZ) or both HBZ and Tax (HMLΔ-Gal4; UAS-Tax>UAS-HBZ) (n = 30, from three independent crosses). Data represent mean ± standard error of the mean. p< 0.0001 (****). **(E, F)** Levels of expression of Relish and Diptericin in the control (GMR-GAL4>*w1118)*, Tax Tg (GMR-Gal4>UAS-Tax), HBZ Tg (GMR-Gal4>UAS-HBZ), or Tax/HBZ Tg (GMR-Gal4;UAS-Tax>UAS-HBZ) transgenic adult flies heads. The values were normalized to Rp49. Reported values are the average of three independent experiments and error bars represent SD of triplicates p<0.01 (**) p<0.001 (***). **(G)** Cell lysates (150 μg) from control

(GMR-GAL4>*w1118*), Tax Tg (GMR-Gal4>UAS-Tax), HBZ Tg (GMR-GAL4>UAS-HBZ) or Tax/HBZ Tg (GMR-Gal4; UAS-Tax>UAS-HBZ), transgenic adult flies heads were analyzed by western blotting and probed with indicated antibodies.

transgenics, the co-expression of Tax and HBZ in flies did not yield any apparent rough eye phenotype (Fig 3B and 3C). This suggests that overexpression of HBZ totally reversed Tax-mediated rough eye phenotype. This result was further asserted while testing the effect of HBZ expression on the number of circulating hemocytes, in the third instar larvae of the double *tax-hbz* transgenic flies. While Tax expression induced a significant increase in the number of circulating hemocytes, co-expression of Tax and HBZ yielded similar hemocyte numbers as those seen in control or *hbz* transgenic flies (Fig 3D). This result is consistent with the lack of ommatidial transformation in the double transgenic fly model.

At the functional level, transcripts of Relish and Diptericin were significantly reduced in the double *tax/hbz* transgenic flies as compared to *tax* transgenics (Fig 3E and 3F). Importantly, a significant decrease in E(z), SUZ12 and H3K27me3 level was observed in the double transgenic flies as compared to *tax* flies (Fig 3G). These results demonstrate that HBZ expression in *tax* transgenic flies significantly attenuates Tax-mediated NF-κB activation, PRC2 activation and *in vivo* transformation.

## HBZ overexpression in *tax* transgenic flies alleviates Tax-induced senescence *in vivo*

HBZ has been shown to alleviate Tax-induced senescence resulting from activation of the NF-kB pathway *in vitro* [16,17]. In accordance with these findings, our *in vivo* fly model revealed that Tax expression in larvae from *tax* transgenic flies was associated with induction of senescence (Fig 4A). This effect was fully reverted by the concurrent expression of HBZ in larvae of the double *tax-hbz* transgenics (Fig 4A). To confirm these results, we used Dacapo, a *Drosophila* cyclin-dependent kinase inhibitor and a p21/p27 homologue considered as a marker of cell cycle arrest [48]. The amount of *dacopo* mRNA was significantly increased in *tax* flies compared to control flies (Fig 4B). This effect was fully reverted in *tax-hbz* flies demonstrating that HBZ interferes with Tax-induced senescence following excessive activation of NF-κB. We further confirmed these results in the *Drosophila* hematopoietic system. Tax and HBZ expression in larvae were confirmed by qRT-PCR (Fig 4C). Third instar larvae from *tax* transgenics showed significantly increased SA-β-gal activity in hemocytes as compared to control larvae (Fig 4D and 4E). Interestingly, the co-expression of Tax and HBZ totally reverted this phenotype (Fig 4D and 4E). These results strongly support the *in vitro* data [16,17] and clearly demonstrate that HBZ alleviates Tax-induced senescence *in vivo*.

Finally, to address the importance of the PRC2 complex in HBZ-mediated inhibition of Tax-mediated NF-κB activation and senescence, we assessed Tax-induced senescence in Tax/ Relish RNAi (defective for NF-κB pathway), Tax/E(z) RNAi and Tax/SUZ12 RNAi (defective for PRC2 pathway), and used Tax/mcherry RNAi as a control. Third instar larvae from Tax/ Relish RNAi transgenics showed a sharp decrease in SA-β-gal activity in their hemocytes, as compared to control larvae (Fig 4F), demonstrating the disappearance of senescence phenotype in absence of Tax-mediated NF-κB activation. Importantly, a partial but significant rescue of Tax-mediated NF-κB and its subsequent induced senescence were obtained in Tax/E(z) RNAi and Tax/SUZ12 RNAi Transgenics, when compared to the control flies (Fig 4F). Altogether, these results implicate PRC2 core components E(z) and SUZ12, at least partially, in Tax-mediated NF-κB activation and senescence, and prove a reciprocal link between these two key pathways in Tax-mediated transformation.

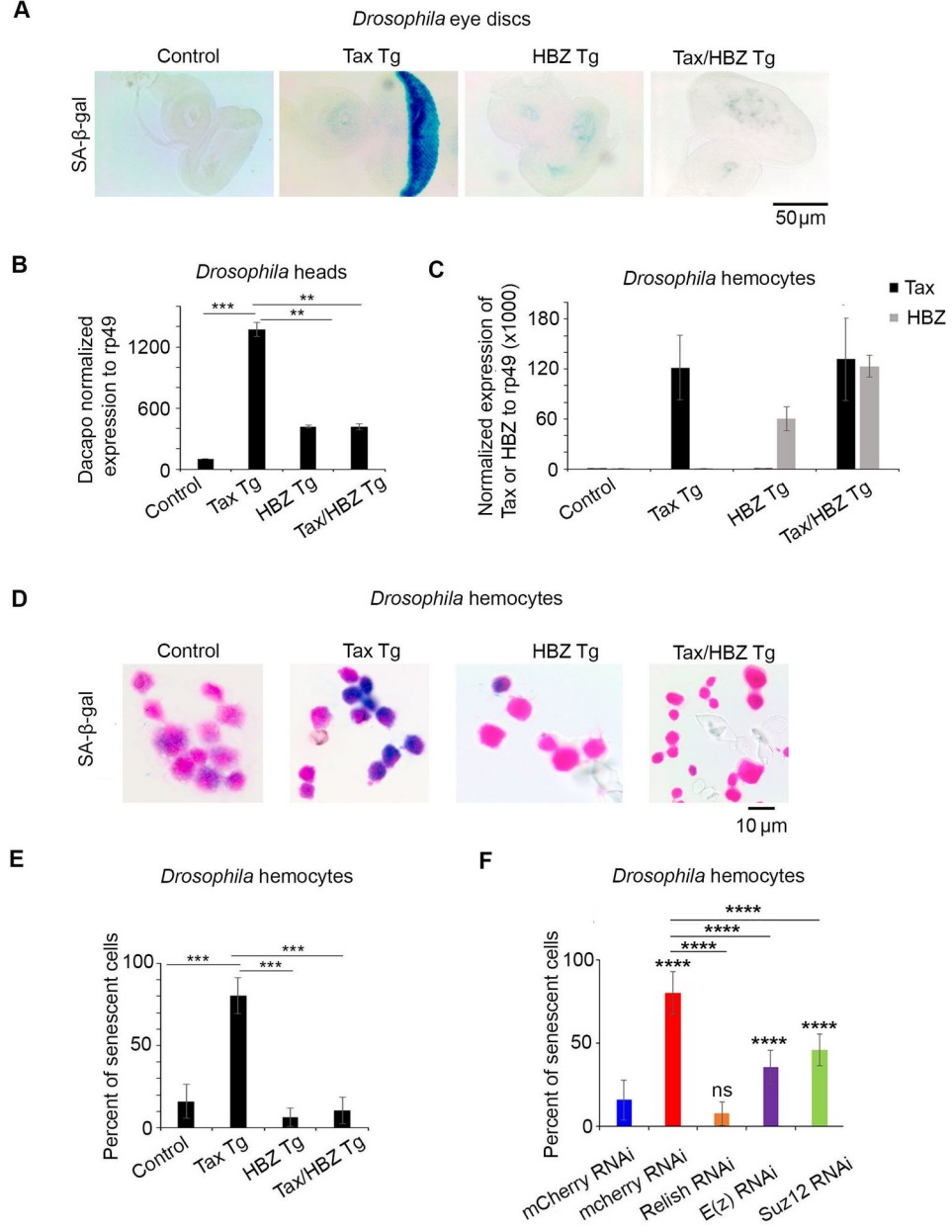

**Fig 4. HBZ overexpression alleviates Tax-induced senescence *in vivo*. (A)** SA-β-gal expression in larval eye-imaginal discs from control (GMR-GAL4>*w1118)*, Tax Tg (GMR-Gal4>UAS-Tax), HBZ Tg (GMR-Gal4>UAS-HBZ) or Tax/HBZ Tg (GMR-Gal4; UAS-Tax>UAS-HBZ). **(B)** Levels of expression of Dacapo (p21/p27) in the transgenic adult flies heads as indicated. The values were normalized to Rp49. Reported values are the average of three independent experiments and error bars represent SD of triplicates p<0.01 (**) p<0.001 (***). **(C)** Levels of expression of Tax and HBZ transcript levels in control larvae (HMLΔ-Gal4>*w1118*), larvae expressing transgenic Tax alone (HMLΔ-Gal4>UAS-Tax), HBZ alone (HMLΔ-Gal4>UAS-HBZ) or both HBZ and Tax (HMLΔ-Gal4; UAS-Tax>UAS-HBZ). **(D)** SA-β-gal expression in circulating hemocytes in transgenic larvae as indicated. **(E)** Quantification of senescent hemocytes. Results are expressed as percentage of control. p<0.001 (***). **(F)** SA-β-gal expression in circulating hemocytes in larvae from the control mcherry RNAi (HMLΔ-Gal4> mCherry RNAi), control Tax/mCherry RNAi (HMLΔ-Gal4;UAS-Tax>mCherry RNAi), Tax/Relish RNAi (HMLΔ-Gal4;UAS Tax>Relish RNAi), Tax/E(z) RNAi (HMLΔ-Gal4;UAS-Tax>E(z) RNAi) and Tax/Suz12 RNAi (HMLΔ-Gal4;UAS-Tax>Suz12 RNAi) as indicated. Results are expressed as percentage of control. (ns = not significant), p<0.0001 (****).

## HBZ blocks Tax-induced PRC2 activation in mammalian cell systems

To validate our results generated in *Drosophila*, we overexpressed HBZ, at increasing concentrations, in the mammalian HEK293T cells, transiently expressing Tax. The increasing concentrations of HBZ protein were verified by western blot using an HBZ-myc tag (Fig 5A). In line with previous reports [28], overexpression of Tax resulted in increased EZH2 and SUZ12 levels leading to a global increase of H3K27me3 (Figs 5A and S3). However, co-expression of Tax and HBZ in human cells dramatically decreased the global amount of H3K27me3, as compared to human cells expressing Tax alone (Fig 5A).

To validate these results in CD4+ cells, we transfected Jurkat cells, with either an empty vector (control), His-Tax alone, Myc-HBZ alone, or with His-Tax/Myc-HBZ vectors [49], and assessed the effect of Tax and HBZ viral protein on EZH2 and SUZ12 PRC2 core components and their respective downstream H3K27me3. Consistent with our results in both *Drosophila* and transfected-293T cells, expression of Tax alone yielded an upregulation of EZH2 and SUZ12 protein levels, which coincided with increased H3K27me3 levels (Fig 5B). Expression of HBZ upregulated SUZ12 but not EZH2 protein levels, and also resulted in increased H3K27me3 levels (Fig 5B). Importantly, co-expression of both Tax and HBZ decreased the global amount of H3K27me3 as compared to Jurkat cells expressing either viral proteins (Fig 5B), further validating the fly results in a human CD4+ T cell system.

Based on these observations, we hypothesized that HBZ may affect the enrichment of H3K27me3 at the promoter of Tax-dependent genes. HEK-293T cells were co-transfected with either an empty vector (control) or HIS-Tax, alone or together with Myc-HBZ expression plasmids. 48 hours post-transfection, cells were processed by chromatin immunoprecipitation (ChIP)-qPCR, to analyse the effect of HBZ expression on the enrichment of H3K27me3 at gene promoters. As previously reported, transfection of HEK293T cells with Tax expression vector significantly increased H3K27me3 accumulation at the promoter of Tax target genes such as CDKN1A, NDRG2, and HEG1 as compared to untransfected cells (Fig 5C). Co-transfection of Tax and HBZ reduced the amount of H3K27me3 at these promoter regions, as compared to cells transfected with Tax alone. As expected, the overexpression of HBZ alone did not induce any significant modification in H3K27me3 levels at these promoters as compared to un-transfected cells since they are Tax-dependent (Fig 5C). Consistent with the H3K27me3 enrichment, CDKN1A and NDRG2 transcript levels were lower in Tax expressing cells and their levels were restored upon co-expression of HBZ in Tax transfected cells (Fig 5D). We further tested the specificity of our results and examined the enrichment of H3k27me3 at the promoter region of BIM, an HBZ controlled gene [50] and demonstrated a selective and specific enrichment of H3K27me3 in HBZ expressing cells (Fig 5C). This increase was sustained upon co-expression of Tax and HBZ (Fig 5C). Consistently, BIM transcript levels were lower in HBZ and Tax/HBZ, but not Tax expressing cells (Fig 5D). Altogether, these results show the specificity of our assay on both Tax- and HBZ-dependent genes.

To further investigate the effect of HBZ on PRC2 activity in ATL-derived cells, we knocked down HBZ in MT-1 cells. These cells are known to express HBZ with undetectable Tax protein [21,37]. HBZ knockdown (Fig 5E) led to a significant increase in both EZH2 and H3K27me3 levels whereas SUZ12 level was not affected (Fig 5E). Immunoprecipitation experiments confirmed Tax interaction with EZH2 (Fig 5G). Interestingly, co-transfection of Tax and HBZ abolished the interaction between EZH2 and Tax (Fig 5G), pointing out a potential competition between HBZ and Tax for EZH2 binding.

## HBZ protein interacts with PRC2 core components

We then assessed the effect of HBZ on the expression of PRC2 complex components in human cells. Similar to the results generated in the *hbz* fly model, increased SUZ12 and H3K27me3

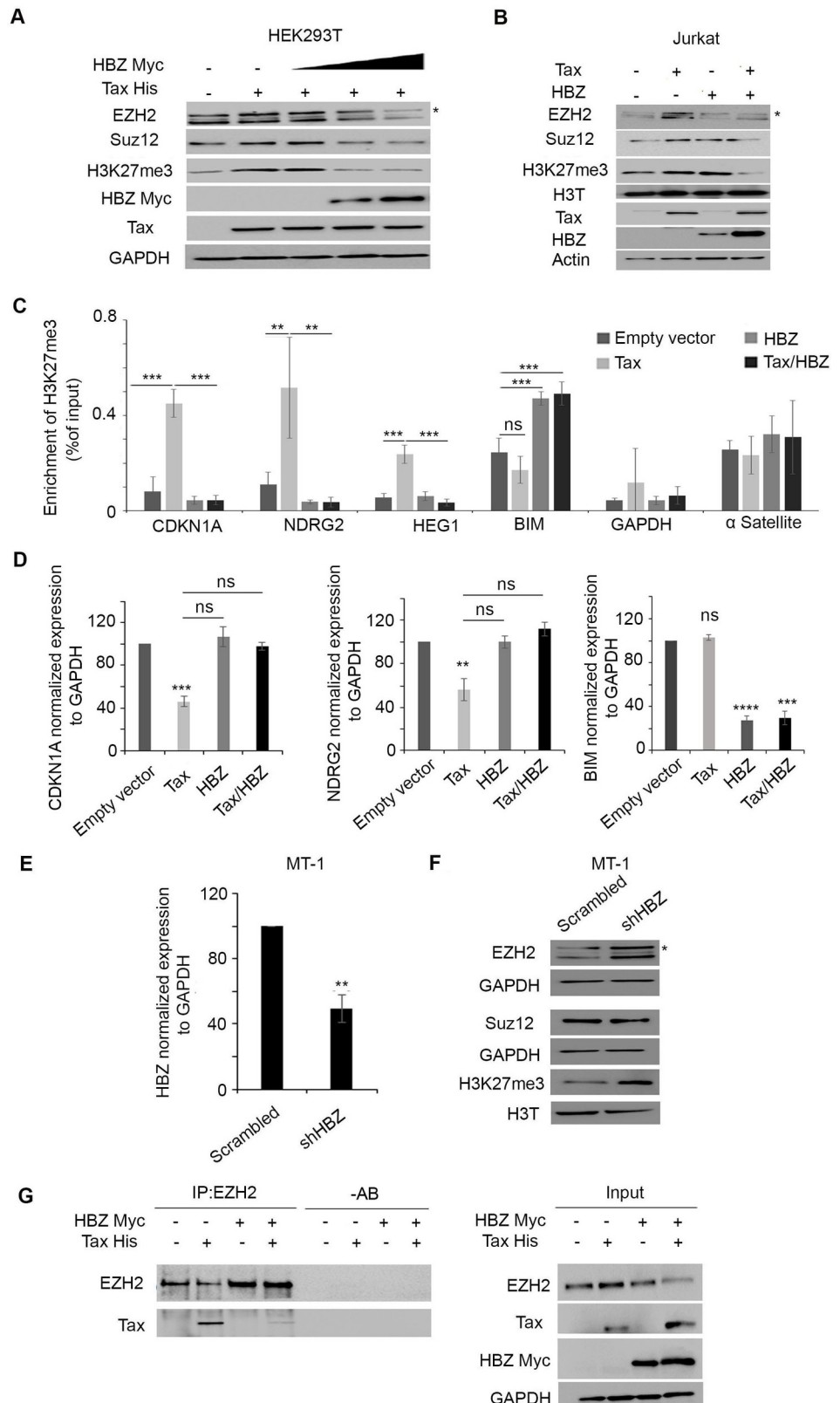

**Fig 5. HBZ overexpression prevents Tax-induced PRC2 activation in human cells. (A)** HEK293T cells were transiently co-transfected with His-Tax and increasing amount Myc-HBZ. Western blot was performed with indicated

antibodies. Asterisk denotes to the band corresponding to EZH2. **(B)** Jurkat cells were transiently co-transfected with an empty vector (control), Tax, HBZ, or Tax and HBZ expression plasmids. Western blot was performed with indicated antibodies. Asterisk denotes to the band. **(C)** Chromatin immunoprecipitation (ChIP)-qPCR analyses were performed on HEK293T cells co-transfected with either an empty vector or HIS-Tax alone or together with Myc-HBZ. 48 hours post-transfection, genomic DNA was fixed with 1% formaldehyde and sheared using Bioruptor. Sheared chromatin was diluted and immunoprecipitated using anti-H3k27me3 antibody. GAPDH was used as a negative control and α-satellite as a positive control. The immunoprecipitated material was quantified by qPCR. Results were normalized to inputs and expressed as %DNA input. (ns = not significant), p<0.001 (***) and p<0.01 (**). **(D)** Transcript levels of CDKN1A, NDRG2 and BIM. Reported values are the average of three independent experiments, and the error bars represent SD of the triplicates. (ns = not significant), p<0.01 (**).), p<0.001 (***), p<0.0001 (****). **(E)** MT-1 cells were transduced with non-targeting control shRNA (sh scrambled) or shRNA against HBZ (shHBZ). qRT-PCR of HBZ confirming HBZ knockdown in MT-1 cells. Reported values are the average of three independent experiments, and the error bars represent SD of the triplicates. p<0.01 (**). **(F)** Western blot analysis was performed with indicated antibodies in MT-1 cells transduced with shHBZ or sh Scrambled. Asterisk denotes to the band corresponding to EZH2. **(G)** HEK293T cells transiently transfected with an empty vector (control), Tax, HBZ, or Tax and HBZ expression plasmids. EZH2 immunoprecipitates (IP: EZH2) were blotted against EZH2 and Tax (Left panel).–AB represents a control with no antibody. Corresponding control cell lysates are shown as indicated (Right panel).

amounts were observed in HEK293T cells transfected with *hbz* expressing vectors (Fig 6A). Furthermore, HBZ physically interacts with endogenous EZH2 and SUZ12 in *hbz* transfected 293T cells (Fig 6B). These results were confirmed using the proximity ligation assay, whereby endogenous EZH2 and SUZ12 interact with exogenous HBZ in the nucleus of HeLa cells (Fig 6C), as well as with endogenous HBZ in MT-1 cells (Fig 6C). Yet, HBZ only partially co-localizes with EZH2 in discrete nuclear foci (Fig 6D). In contrast, the co-localization between HBZ and SUZ12 was observed in speckle-like structures in the nuclear region (Fig 6D). Altogether, these results validate our *in vivo* findings in *hbz* transgenic flies and reveal that HBZ interacts with the PRC2 components EZH2 and SUZ12 in mammalian cell systems and ATL-derived cells.

## Discussion

In the current report, we established an *in vivo hbz* transgenic fly model, and demonstrated that, unlike Tax, HBZ did not induce neither transformation nor NF-κB activation. Using our previously established *tax* transgenic fly model, we confirmed that Tax activates the PRC2 complex *in vivo*. This finding is in line with previously reported data [28]. Despite HBZ failure to induce cellular transformation, we found that HBZ directly interacts with two components of the PRC2 complex, EZH2 and SUZ12, leading to PRC2 activation and H3K27me3 accumulation. Interestingly, overexpression of HBZ in *tax* flies abrogated Tax-induced NF-κB activation and its subsequent senescence, inhibited Tax-induced PRC2 activation and totally rescued the transformation phenotype.

HTLV-1 pathogenicity is linked to both Tax and HBZ [51,52,53]. The oncogenic capacity of Tax was extensively studied and proven *in vitro* and *in vivo* [5–9]. Indeed, Tax expression in rat fibroblasts, T-lymphocytes, or peripheral blood mononuclear cells increased cell proliferation and resulted in cellular transformation and immortalization [28,54,55].

Although Tax protein is undetectable in most primary ATL cells, long-term survival of the bulk of ATL cells may depend on transient bursts of Tax expression in some, if not the majority, of ATL cells and/or in HTLV-1 infected non-malignant cells [19,21]. Conversely, HBZ is constantly expressed in almost all ATL cells suggesting a critical role for HBZ in the ATL phenotype. While silencing of Tax results in apoptosis and cell death [20], silencing of HBZ decreased proliferation of ATL cells but did not result in cell death *in vitro* [38]. *In vivo*, Tax expression yielded leukemia with ATL-like features and NF-κB activation in transgenic mice [5,6,56]. HBZ oncogenic potential *in vivo* was documented in two transgenic mouse models.

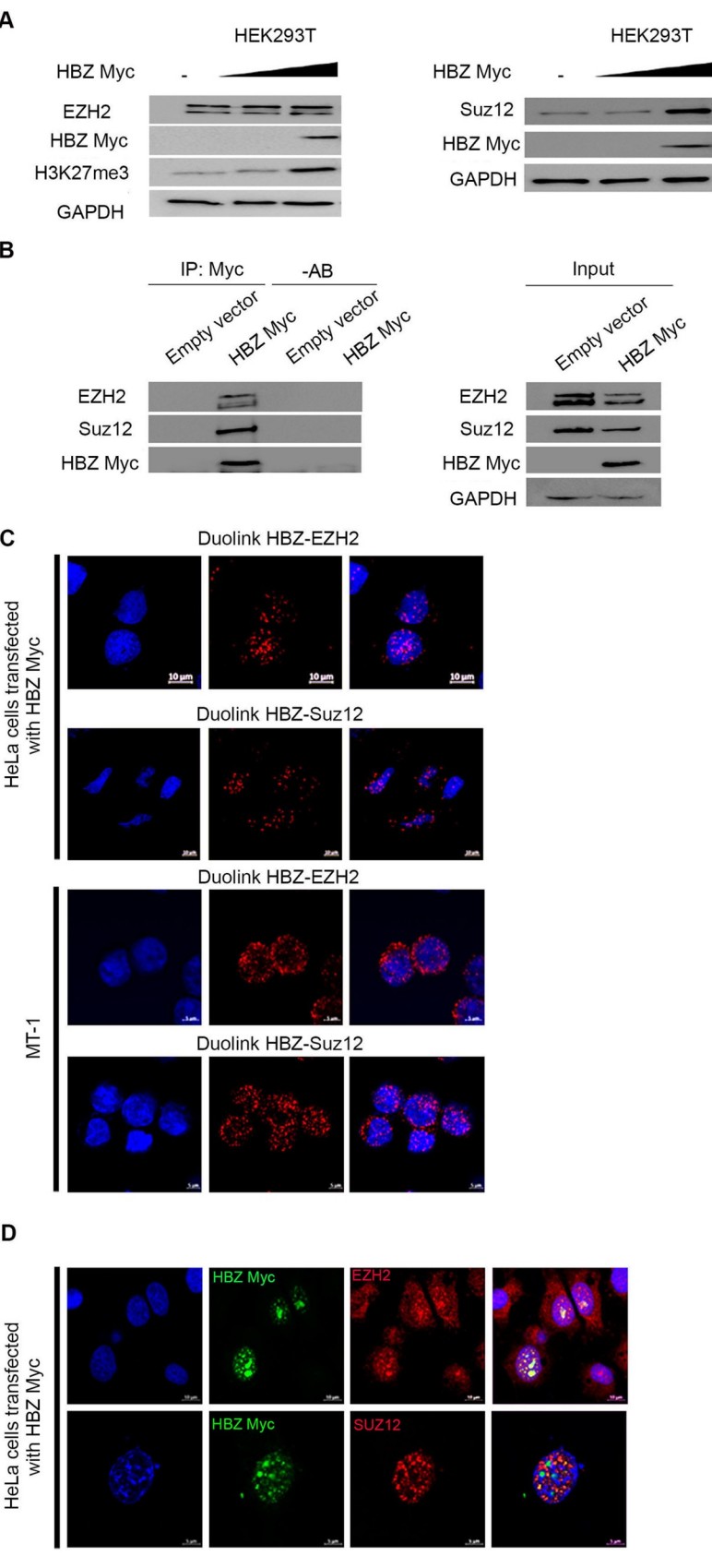

**Fig 6. HBZ interacts with PRC2 components EZH2 and SUZ12 and is associated with H3K27me3 accumulation in human cells. (A)** HEK293T cells transiently transfected with Myc-HBZ. Western blot was performed with indicated antibodies. **(B)** HEK293T cells transiently transfected with Myc-HBZ. Myc immunoprecipitates (IP: Myc) were blotted against EZH2, SUZ12 and Tax (Left panel).−AB represents a control with no antibody. Corresponding control cell lysates are shown as indicated (Right panel). **(C)** HBZ-EZH2 and HBZ-SUZ-12 interactions detected by Duolink Proximity Ligation [73] assay in HeLa cells transfected with HBZ (Scale bar 10 μm), or MT-1 expressing endogenous HBZ. Nuclei were stained with 4,6 diamidino-2-phenylindole (DAPI) (blue). (Scale bar 5 μm). **(D)** Confocal microscopy (Z-Stacks) of HeLa cells transiently expressing Myc-HBZ, and showing partial co-localization of HBZ-EZH2 (Upper panel) and HBZ-SUZ12 (Lower panel). Nuclei were stained with 4,6 diamidino-2-phenylindole (DAPI) (blue) (Scale bars 10μm and 5μm respectively).

The first one predominantly showed inflammatory changes with lymphoma-like changes in some mice [57]. The second one, using Granzyme B promoter, revealed a delayed lymphoproliferative disease in two-thirds of mice [58]. Yet, no evidence was observed in any of these *hbz* transgenics pertaining to NF-κB activation, a hallmark of ATL [59,60]. Recently, a humanized mouse model infected with HTLV-1 lacking functional HBZ showed a similar lymphoproliferative disease with no survival difference compared to wild type HTLV-1, suggesting that HBZ is not essential for tumor development *in vivo* [61]. Ideally, these mice models are potentially more relevant than flies to study the biology of HTLV-1, known to infect primates. Unfortunately, a main drawback in these murine models is the very long latency period, which exceeds 18 months for *tax* transgenics [6], *hbz* transgenics [58] or *tax/hbz* double transgenics [62]. This long latency may be attributed to the acquisition of cellular somatic mutations, which makes it impossible to distinguish whether an observed phenotype or change in cell biology is attributed to the viral oncoprotein or to the acquired somatic mutations.

Many advantages pertain to the use of *Drosophila* models in that setting. These include the fast generation time, the high number of progeny, and the availability of easy genetic screens. Indeed, a nearly complete collection of mutants, RNAi lines, and a high number of driver systems (which allow for tangible expression of phenotypes in different sites including eyes and hemocytes), are accessible. Furthermore, key signaling pathways are highly conserved between mammals and flies, including the NF-κB pathway [46], the PRC2 complex [40] and many other signaling pathways regulating blood cell differentiation [63,64]. In this *Drosophila* model, Tax expression induced cellular transformation, NF-κB activation and increased hemocyte count [10]. Here we show that HBZ expression failed to induce cellular transformation and only resulted in minimal increase in hemocyte count. Yet, HBZ alone upregulated SUZ12 expression and activated the PRC2 complex. Surprisingly, overexpressing HBZ in *tax* flies, abrogated eye roughness and abolished the prominent increase in hemocyte count in *tax* transgenics demonstrating an antagonistic effect of HBZ on Tax-mediated transformation *in vivo*. Yet, these interesting *in vivo* findings in a *Drosophila* model still require further validation in mammalian systems.

Tax is an immunogenic viral protein [65,66], and its downregulation is a fundamental step to evade the host immune system, and allow ATL development [67]. Tax is also a potent activator of the NF-κB pathway [59,60,68], even if the activation of this pathway can be also due to different host factors such as the high frequency of activation mutation in TCR signaling pathway [25] and the loss of the microRNA (mir31) [69]. In *tax* transgenic flies, Diptericin expression was increased, indicative of NF-κB activation [10]. This induced persistent and constitutive NF-κB activation triggers Tax-induced senescence [17,70]. Indeed, upon NF-κB activation, P65/RelA deregulates p21/p27 checkpoint leading to cellular senescence [17]. In line with this concept, we demonstrated that Tax expression in flies induced *in vivo* senescence in hemocytes and larvae imaginal disks. HBZ leads to P65 degradation, thus inhibiting NF-κB [32]. In agreement with these results, *hbz* transgenic flies did not exhibit any increase in relish or its downstream Diptericin expression, concurrent with the absence of the eye roughness

phenotype. Interestingly, overexpressing HBZ in *tax* transgenic flies abrogated Tax-induced NF-κB activation *in vivo*. This is consistent with a previous report where HBZ downregulated NF-κB activity in Hela cells transiently expressing Tax protein [17] and relieved Tax-imposed senescence, hence allowing cell proliferation and persistent infection [17]. Here, we confirmed these results *in vivo*, and demonstrated that HBZ expression in *tax* transgenic flies alleviated senescence and decreased Dacapo (P21/p27) expression. The difference in Tax and HBZ transformative potential may be due, at least in part, to an antagonistic capacity in NF-κB modulation. In that sense, HBZ alone cannot transform cells but it may function to alleviate the deleterious effects of Tax on cells, hence allowing persistent infection, cell proliferation and subsequent transformation. Nevertheless, HBZ is capable of maintaining PRC2 activation in the absence of Tax.

Epigenetic alterations play a critical role in ATL development. Indeed, primary ATL cells harbor a deregulated PRC2 activity with overexpression of EZH2 and increased H3k27me3 levels [71]. In ATL cells, an ATL-specific epigenetic landscape comprising PRC2 hyper-activation with genome-wide accumulation of H3K27me3 repressive mark at half of genes was reported [28]. Tax binds PRC2 components and increases the transcription of EZH2 and SUZ12 resulting in PRC2 activation and H3k27me3 accumulation [28]. Consistent with these findings, we observed an increased expression of EZH2, SUZ12 and H3k27me3 accumulation in cells and flies overexpressing Tax. More importantly, Tax interacted with EZH2 and SUZ12 and increased recruitment of H3k27me3 repressive marks to the promoters of genes reported to be deregulated in ATL [28] such as the tumor suppressor genes CDKN1A and NDRG2 whose expression is downregulated in ATL. Unlike relish silencing that totally rescued eye roughness and hemocyte count, silencing EZH2 and SUZ12 expression in *tax* transgenic flies only resulted in a partial rescue of Tax-induced eye roughness and hemocyte count. Our results implicate PRC2 core components E(z) and SUZ12 in Tax-mediated cell proliferation, NF-κB activation and senescence. Similarly, Tax-induced NF-κB activation enhanced PRC2 activity hence creating an activation loop.

Despite its ability to increase PRC2 activity and H3k27me3 accumulation, HBZ expression in flies did not induce *in vivo* transformation, suggesting that PRC2 complex activation is necessary but not sufficient for cellular transformation. This HBZ-mediated increase in PRC2 activity and H3k27me3 accumulation is consistent with the reported deposition of H3K27me3 at the promoter of *bim*, a target gene of HBZ [72]. Therefore, it appears that HBZ, like Tax, is capable of affecting epigenetic pathways in HTLV-1 infected cells. In line with previous reports [28], HBZ didn't affect the expression of EZH2. Yet, HBZ co-localized and interacted with EZH2 and SUZ12. Surprisingly, HBZ expression abolished Tax-mediated PRC2 activation in both *tax/hbz* flies and Tax/HBZ expressing cells. To explain these opposite HBZ effects on PRC2 activation, our hypothesis is that, in the absence of Tax, HBZ activates the PRC2 activity, likely through upregulation of SUZ12 and direct interaction with EZH2. Conversely, HBZ suppresses Tax-induced PRC2 activation, likely because of inhibition of Tax-induced NF-κB activation, resulting in inhibition of Tax-induced upregulation of EZH2 protein levels, as well as possibly through competitive binding of Tax and HBZ with components of the PRC2 complex. Indeed, silencing HBZ in MT-1 cells with sporadic burst of Tax expression resulted in increased EZH2 and H3k27me3 levels. Whether Tax and HBZ interact with the same or different domains of EZH2 remain to be explored. Finally, based on the importance of NF-κB and PRC2 activation in Tax-induced transformation, NF-κB and PRC2 inhibition in Tax/HBZ co-expressing cells contributes to the reversal of Tax effects.

In conclusion and based on the generated data in *Drosophila* transgenic models, we hypothesized that Tax and HBZ play important roles in HTLV-I transformation and persistence (Fig 7). Bursts of Tax expression would lead to PRC2 activation, hyper-activation of NF-κB,

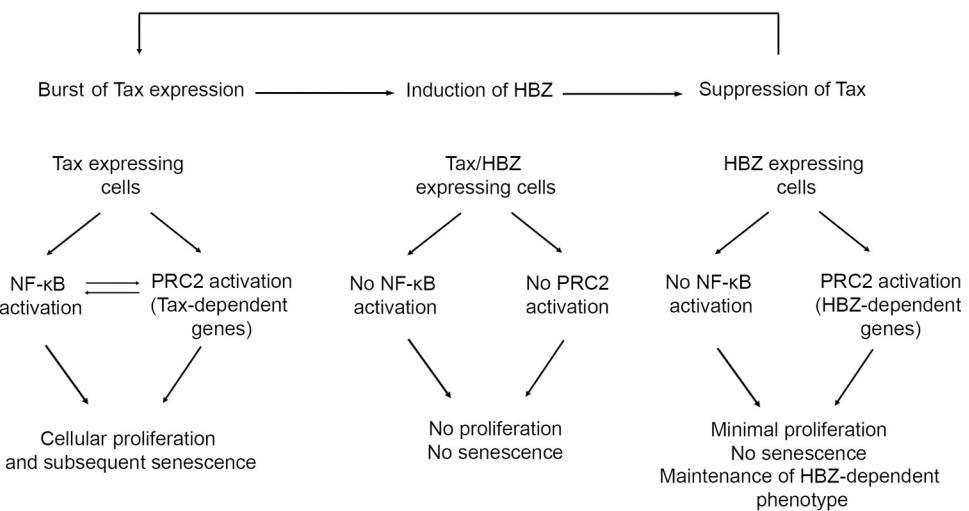

**Fig 7. Proposed model.**

excessive proliferation, senescence and immune activation. HBZ antagonizes these Tax delete-rious effects, preventing hyper-activation of NF-κB and senescence, hence allowing long-term persistence of infected cells. HBZ expression leads to Tax silencing [36]. In the absence of Tax, HBZ expression will maintain PRC2 activation. However, transient bursts of Tax expression will occur again [19,21] and serve for keeping basal NF-κB activation, and allowing genetic instability and accumulation of somatic mutations. These will eventually result in full ATL phenotype in some infected individuals. Again, these results demonstrated in a fly system will require further validation in a mammalian cell system.

Overall, and despite a number of limitations in a non-mammalian system, we report a pow-erful *in vivo* tool to investigate the antagonistic effects of HBZ on Tax-induced transformation, cellular effects and epigenetic modulation.

## Materials and methods

### Fly stocks

The UAS-myc-*Tax* [10], the Oregon-R w[1118] line (referred to as wild-type), *Relish*-RNAi (VDRC#49414), *E(z)*-RNAi (BDSC#33659), *SUZ12*-RNAi (BDSC#33402), GMR-GAL4 (BDSC #1104), and hemocyte-specific drive Hemolectin-GAL4 delta (BDSC#30139) *Drosophila* stocks were grown on standard cornmeal yeast extract medium at 25°C.

### Generation of *hbz* transgenic flies

*hbz* transgenic flies were generated using the Phi C31 integrase system and were inserted on the 2[nd] chromosome for UAS-Gal4 expression. *hbz-myc-His* was inserted into the pUAST-attB *Drosophila* expression vector (Custom DNA cloning). pUAST- attB-myc-his *hbz* was then injected into y1 w67c23; P{CaryP ABLattP2 (8622 BDSC) embryos to generate the transgenic flies (BestGene Inc,Chino Hills, CA).

### Scanning electron microscopy (SEM)

Scanning electron microscopy experiments were performed according to previously described procedures [10,42]. In brief, adult flies were fixed in 2% glutaraldehyde/ 2% formaldehyde/

PBS solution. Flies were then dehydrated, by sequential washes of increased ethanol concentrations, dried using a critical point dryer (k850, Quorum Technologies), mounted with standard aluminum heads, and finally coated on 20nm gold layer. Images were obtained using the Tescan, Mira III LMU, Field Emission Gun (FEG) SEM with Secondary Electron detector.

## Scoring of eye phenotypes

Eye phenotypes were scored utilizing a previously established grading score by our group [10,42]. This scoring system takes into consideration the degree of bristle organization, ommatidial fusions, and loss of ommatidia. SEM images were coded by a researcher and blindly analyzed by another. Graphs represent the analysis of three independent crosses, where 15–20 flies from each cross were scored and classified into four phenotypic classes.

## Hemocyte count

*Wild type*, *tax*, and *hbz* flies were crossed to the Hemolectin-Gal4 delta (HMLΔ-Gal4) hemocyte driver. Third instar larvae of the indicated genotypes were bled into PBS (Sigma) and hemocytes were counted as described [10]. Hemocytes from thirty larvae of each genotype were bled and counted, from three independent crosses. Statistical analysis was performed using Student's t-test P < 0.05.

## Eye disc and hemocyte *β*-galactosidase assay

For eye disc senescence analysis, third instar larvae were dissected in PBS, kept on ice, fixed in 0.2% glutaraldehyde/2% formaldehyde/PBS at room temperature for 5 min, and rinsed three times with PBS. Larvae were then incubated in staining solution (1x PBS pH7.5, 1mM MgCl2, 4mM potassium ferricyanide, 4mM potassium ferrocyanide, 1% Triton, 2.7mg/ml X-Gal) overnight at 37˚C. Stained larvae were washed twice with PBS and eye imaginal discs were dissected and mounted on slides using a Prolong Anti-fade kit (Invitrogen, P36930). Images were taken by light microscopy using Olympus CX41 Microscope.

For hemocyte senescence analysis, third instar larvae of the indicated genotypes were bled into PBS (Sigma) and hemocytes were fixed in 0.2% glutaraldehyde/2% formaldehyde/PBS at room temperature for 5 min, rinsed three times with PBS, then incubated in staining solution (1x PBS pH7.5, 1mM MgCl2, 4mM potassium ferricyanide (Sigma), 4mM potassium ferrocyanide (Sigma), 1% Triton, 2.7mg/ml X-Gal). Following overnight incubation at 37˚C, hemocytes were washed twice with PBS, and the percentage of positively stained hemocytes was determined after counting 10 random fields to reach 100 hemocytes.

## Cell culture

HEK293T and HeLa cells were maintained in Dulbecco's Modified Eagle's Medium (DMEM, GIBCO), supplemented with 10% fetal bovine serum (FBS, GIBCO). ATL-derived MT-1 cells [gift from K. Ishitsuka] were maintained in RPMI medium (Sigma) supplemented with 2 mM L-glutamine (Sigma), 10% fetal calf serum (Sigma) and antibiotics.

## Transient transfection and transduction

pcDNA3.1-*hbz*-Myc-His and pcDNA3.1-*Tax*-His expression vectors were kindly provided by R. Mahieux (Journo et al., 2013; Dubuisson et al., 2018). Plasmid transfections were performed using Lipofectamine 2000 (Gibco, Invitrogen). Jurkat cells were transfected using the Amaxa cell line Nucleofector V kit (Lonza) program X-001, according to the manufacturer's instructions. MT-1 cells were transduced using green fluorescent protein (GFP)-lentiviral vectors

encoding scrambled (SCR) or shRNA against HBZ (kindly provided by M. Matsuoka) (Satou et al., 2006). Lentiviruses were produced by transient transfections of HEK-293T cells. Infection of target cells was performed by spinoculation for 3h at 1500 rpm and at 32˚C.

## Quantitative PCR

Total RNA was extracted with TRIzol (Qiagen) following the manufacturer's instructions. RNA was DNAse-treated (Turbo DNA-free AM1907, Ambion). One µg of RNA was reverse transcribed using iScript III (Biorad). Syber green qRT PCR was performed using the BIORAD CFX96 machine. Primers for D-jun, Tax, HBZ, Relish and Diptericin are listed in S1 Table. Individual reactions were prepared using 0.25 µM of each primer, 150 ng of cDNA and the SYBR Green PCR Master Mix in a final volume of 10 µl. PCR reaction consisted of a DNA denaturation step at 95˚C for 3 min, followed by 35 cycles (denaturation at 95˚C for 15 sec, annealing at 57˚C for 60 sec, extension at 72˚C for 30 sec). Expression of individual genes was normalized to the housekeeping gene Glyceraldehyde-3-Phosphate dehydrogenase *gapdh* for mammalian cells and *ribosomal protein 49 (Rp49)* for *Drosophil*a extracts (S1 Table). The transcript expression level was calculated according to the Livak method (Livak and Schmittgen, 2001). Each experiment was performed using duplicates from three biological independent experiments.

## Antibodies

The following primary antibodies were used in western blot, immunofluorescence or chromatin immunoprecipitation assays as indicated: mouse anti-c-Myc (ThermoFisher cat#9E10), mouse anti-His (Santa Cruz cat#sc57598), rabbit anti-EZH2 (Invitrogen cat#36–6300), rabbit anti-SUZ12 (Cell signaling cat#D39F6), rabbit anti-H3k27me3 (Active motif Cat#39155), rabbit anti-H3k4me3 (EMD Millipore cat#07–473), rabbit anti-βactin (Abcam cat#ab8227), rabbit anti-Histone H3 (Abcam cat#ab1791), mouse anti-Tax (cat#168-A51 from the National Institutes of Health AIDS Research and Reference Reagent Program), mouse anti-HBZ (a gift from J.M. Péloponèse), rabbit anti-E(z), rabbit anti-SUZ12 (a gift from J. Müller), mouse anti-Relish (DSHB cat#C21F3) and anti-GAPDH (B2534M-HRP Abnova), mouse anti-c-Myc-tag (9E10) (Abcam cat#ab32), rabbit anti-EZH2 (Merck Millipore cat# 07–689), Goat anti-mouse Alexa Fluor-488 (Abcam cat≠150113), Goat anti rabbit -rabbit Alexa Fluor-594 (Abcam cat ≠150080).

## Immunoblot analysis

Fly heads, third instar larvae or cells were homogenized in Laemmli buffer, sonicated (Bioruptorpico, 10 cycles, 30 sec on/off), and washed with PBS. One hundred fifty µg proteins from twenty to thirty flies or one hundred µg proteins from cell lysates were loaded onto a 10% or 12% SDS-polyacrylamide gel, subjected to electrophoresis, and transferred onto nitrocellulose membranes. Membranes were then blocked in 5% non-fat milk and incubated with specific primary antibodies as indicated. Bands were visualized by chemiluminescence (Clarity max, Bio-Rad, Cat# 170–5061).

## Immunofluorescence, *in situ* proximity ligation assays (Duolink), and confocal microscopy

For immunofluorescence assays, HeLa cells were cultured on coverslips and were fixed with 4% paraformaldehyde for 10 min, 48h following different transfections. Cells were permeabilized in PBS-Triton X-100 0.5% for 30 min at room temperature, washed in (PBS/0.05% Triton X-100) buffer and blocked in (1% BSA and 10% goat serum) at room temperature for 1h.

Transfected cells were incubated with anti-c-Myc (200 μg/ml), anti-EZH2 (0.25 mg/mL) or anti-SUZ12 (100 μg/ml) primary antibodies diluted at 1:50, 1:100 and 1:100 in blocking buffer respectively, for 2h at room temperature. Primary antibodies were revealed by Alexa Fluor-488 or 594 labeled secondary antibodies from Abcam. Staining of nuclei was performed using DAPI for 5 min and then coverslips were mounted on slides using a Prolong Anti-fade kit (Invitrogen, P36930).

Protein-protein interactions were visualized using the Duolink *in situ* proximity ligation assay [73] system (Olink Bioscience) following the manufacturer's instructions using anti-HBZ, anti-EZH2, anti SUZ12, anti-His, or anti-c-Myc antibodies.

Images or image-stacks (average thickness 30 μm) were acquired by confocal microscopy using a Zeiss LSM710 confocal microscope (Zeiss, Oberkochen, Germany) with a Plan Apochromat 63/1.4 numeric aperture oil-immersion objective, using Zen 2009 (Carl Zeiss).

## Immunoprecipitation

For immunoprecipitation assays, HEK293T were seeded at the density of two million cells, in 10-cm culture dishes. Cells were harvested 48h post-transfection. Nuclear extracts from HEK293T cells were washed in PBS 1X and lysed in 10 mM Hepes pH 7.9, 10 mM KCl, 1.5 mM MgCl2, 0.5 mM DTT, and EDTA-free protease inhibitor cocktail (Roche cat≠11836145001). After 10 min on ice, lysates were centrifuged at 2100 rpm for 5 min at 4˚C and pellets were re-suspended in 10 mM Hepes pH 7.9, 150 mM NaCl, 5 mM MgCl2, 0.1% NP40 (Sigma cat≠I8896), and EDTA-free protease inhibitor cocktail. The nuclear lysates were incubated for 20 min at room temperature with 0.1 μg/μl DNaseI (Roche). RNase treatment was performed for 15 min at room temperature with 0.1 μg/μl Rnase A (Sigma). Equal quantities of proteins were subjected to immunoprecipitation with the mouse anti-c-myc-tag (9E10) (3μg/ml; ab32), rabbit anti-EZH2 (5μg/ml; Merck Millipore 07–689) antibodies overnight at 4˚C. Formed complexes were linked to magnetic Dynabeads/Protein G (LifeTechnologies) for 4h at 4˚C. After several washes with TBS, 0.5% Tween-20, purified complexes were eluted with Laemmli buffer. Immunoprecipitated proteins and their corresponding lysate controls were analyzed by western blot. Results are representative of two independent experiments.

## Chromatin immunoprecipitation (ChIP)

Around two million HEK293T cells were transfected with 18μg DNA plasmid (9μg pcDNA3.1-Tax-1-His alone or together with 9μg PcDNA3.1-HBZ-Myc-His). Cells were harvested 48hrs later for ChIP assay. In brief, three million cells per IP were cross-linked with 1% formaldehyde for 15 mins at room temperature. Cross-linking was stopped by the addition of 125 mM glycine. Nuclei were isolated using ice cold lysis buffer (50 mM Tris-HCl, pH 8.0, 10 mM EDTA, 1% SDS, 20 mM Na-butyrate (Sigma cat≠B5887), protease inhibitor cocktail (Roche cat≠11836145001). DNA was sheared by sonication using Bioruptor (Diagenode) resulting in fragments between 100 and 400 base pairs. Immunoprecipitation was performed with 5 μg anti-H3K27me3 antibody overnight at 4˚C. After washing with RIPA buffer (10 mM Tris-HCl, pH 7.5, 140 mM NaCl, 1 mM EDTA, 0.5 mM EGTA, 1% Triton X-100, 0.1% SDS, 0.1% Na-deoxycholate (Sigma cat≠D6750) and TE buffer (10 mM Tris-HCl, pH 8.0, 10 mM EDTA), chromatin complexes were eluted with elution buffer (20 mM Tris-HCl, pH 7.5, 5 mM EDTA, 50 mM NaCl, 20 mM Na-butyrate, 1% SDS, 50 μg/ml proteinase K (NEB cat≠P8107S) for 2 hours at 68˚C. washed and purified DNA were subjected to qPCR, using SYBR Green PCR Kit (Biorad) according to the manufacturer's instructions. Primers' sequences are listed in S1 Table. In each condition, the enrichment of protein in the target DNA fragment was compared to the percentage of DNA quantity in the input sample.

## Statistical analysis

Data are presented as mean±SEM. One-way ANOVA with Bonferroni post-hoc test was used to compare means (GraphPad Prism). Differences between means were considered significant when the p-value was less than 0.05.

## Supporting information

**S1 Fig. Validation of Tax and HBZ expression in transgenic *Drosophila* heads and hemocytes. (A)** Levels of expression of Tax and HBZ in the control (GMR-Gal4>*w1118*), Tax transgenics (Tg) (GMR-Gal4>UAS-Tax) and HBZ-Tg (GMR-Gal4>UAS-HBZ) confirming the expression of *tax* and *hbz* transgenes in adult flies heads. Transcript levels were normalized to Rp49. Reported values are the average of three independent experiments and error bars represent SD of triplicates. p<0.01 (**). **(B)** Levels of expression of Tax and HBZ in the control (HMLΔ-Gal4>*w1118*), Tax Tg (HMLΔ-Gal4>UAS-Tax) and HBZ-Tg (HMLΔ-Gal4>UAS-HBZ), confirming the expression of *tax* and *hbz* transgenes in larval hemocytes. Transcript levels were normalized to Rp49. Reported values are the average of three independent experiments and error bars represent SD of triplicates. p<0.1(*).
(TIF)

**S2 Fig. Validation of Tax expression in transgenic *Drosophila* heads and hemocytes RNAi lines. (A)** Cell lysates (150 μg) of transgenic adult flies heads from control (GMR-Gal4>mCherry RNAi), (GMR-Gal4;UAS-Tax>mCherry RNAi), (GMR-Gal4;UAS-Tax>Relish RNAi), (GMR-Gal4;UAS-Tax>E(z) RNAi) and (GMR-Gal4;UAS-Tax>Suz12 RNAi) were analyzed by western blot confirming the expression of *tax* transgene. **(B)** Levels of expression of Relish, E(z), and SUZ12 in the transgenic adult flies heads as indicated. Transcript levels were normalized to Rp49. Reported values are the average of three independent experiments and error bars represent SD of triplicates. p<0.01 (**), p<0.001 (***). **(C)** Cell lysates (300 μg) from control (HMLΔ-Gal4> mCherry RNAi), (HMLΔ-Gal4;UAS-Tax>mCherry RNAi), (HMLΔ-Gal4;UAS Tax>Relish RNAi), (HMLΔ-Gal4;UAS-Tax>E(z) RNAi) and (HMLΔ-Gal4;UAS-Tax>Suz12 RNAi) transgenic larvae were analyzed by western blotting confirming the expression of Tax transgene in larval hemocytes. Indicated genotypes are under the control of the hemocyte-specific promoter (HMLΔ-GAL4).
(TIF)

**S3 Fig. Tax induces activation of PRC2 complex in Human cells.** HEK293T cells were transiently transfected by His-Tax. Western blot was performed with indicated antibodies.
(TIF)

**S1 Table. List of PCR primers used in the study.**
(DOC)

## Acknowledgments

We thank Dr. Margret Shirinian for her comments on the manuscript. We also thank KAS Central Research Science Laboratory (CRSL) at the American University of Beirut for their technical help in scanning electron microscopy imaging. The expert assistance from the DTS basic research core facilities at AUB is appreciated. We thank Muller J for providing antibodies; and Matsuoka M for providing vectors. Stocks obtained from the Bloomington Drosophila Stock Center (NIH P40OD018537) were used in this study.

## Author Contributions

**Conceptualization:** Abdou Akkouche, Sara Moodad, Rita Hleihel, Hiba El Hajj, Ali Bazarbachi.

**Data curation:** Abdou Akkouche, Hiba El Hajj, Ali Bazarbachi.

**Formal analysis:** Abdou Akkouche, Sara Moodad, Hiba El Hajj, Ali Bazarbachi.

**Funding acquisition:** Hiba El Hajj, Ali Bazarbachi.

**Investigation:** Abdou Akkouche, Sara Moodad, Rita Hleihel, Hala Skayneh, Séverine Chambeyron, Hiba El Hajj, Ali Bazarbachi.

**Methodology:** Abdou Akkouche, Sara Moodad, Rita Hleihel, Hala Skayneh, Hiba El Hajj, Ali Bazarbachi.

**Project administration:** Hiba El Hajj, Ali Bazarbachi.

**Resources:** Hiba El Hajj, Ali Bazarbachi.

**Software:** Abdou Akkouche, Ali Bazarbachi.

**Supervision:** Hiba El Hajj, Ali Bazarbachi.

**Validation:** Hiba El Hajj, Ali Bazarbachi.

**Visualization:** Hiba El Hajj, Ali Bazarbachi.

**Writing – original draft:** Abdou Akkouche, Hiba El Hajj, Ali Bazarbachi.

**Writing – review & editing:** Abdou Akkouche, Sara Moodad, Rita Hleihel, Hala Skayneh, Séverine Chambeyron, Hiba El Hajj, Ali Bazarbachi.

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
