## [Decision Letter · Decision Letter 0]

1 Sep 2020

Dear Dr. Bazarbachi,

Thank you very much for submitting your manuscript "In vivo antagonistic role of the Human T-Cell Leukemia Virus Type 1 regulatory proteins Tax and HBZ in oncogenic transformation." for consideration at PLOS Pathogens. As with all papers reviewed by the journal, your manuscript was reviewed by members of the editorial board and by several independent reviewers. In light of the reviews (below this email), we would like to invite the resubmission of a significantly-revised version that takes into account the reviewers' comments.

The reviewers agreed that your findings are of interest and potential importance. However, they also agreed that the manuscript as it stands has certain important limitations. Three points in particular need to be appropriately answered before the paper can be considered for publication. First, you are asked to provide clearer and stronger justification for the use of the Drosophila model, which raises the fundamental question of how relevant the findings are to primates as the natural hosts of the virus. Second, please explain how you reconcile the apparently conflicting actions of HBZ on NF-kappaB and PRC2, either activation or, in the presence of Tax protein, inhibition. Third, two reviewers point out that Tax protein expression is lost in about 50% of cases of adult T-cell leukemia: this observation requires modification of some statements in the text of the paper on the roles of both Tax and HBZ, especially in leukemogenesis.

We cannot make any decision about publication until we have seen the revised manuscript and your response to the reviewers' comments. Your revised manuscript is also likely to be sent to reviewers for further evaluation.

Sincerely,

Charles R. M. Bangham

Associate Editor

PLOS Pathogens

Susan Ross

Section Editor

PLOS Pathogens

Kasturi Haldar

Editor-in-Chief

PLOS Pathogens

orcid.org/0000-0001-5065-158X

Michael Malim

Editor-in-Chief

PLOS Pathogens

orcid.org/0000-0002-7699-2064

The reviewers agreed that your findings are of interest and potential importance. However, they also agreed that the manuscript as it stands has certain important limitations. Three points in particular need to be appropriately answered before the paper can be considered for publication. First, you are asked to provide clearer and stronger justification for the use of the Drosophila model, which raises the fundamental question of how relevant the findings are to primates as the natural hosts of the virus. Second, please explain how you reconcile the apparently conflicting actions of HBZ on NF-kappaB and PRC2, either activation or, in the presence of Tax protein, inhibition. Third, two reviewers point out that Tax protein expression is lost in about 50% of cases of adult T-cell leukemia: this observation requires modification of some statements in the text of the paper on the roles of both Tax and HBZ, especially in leukemogenesis.

Reviewer's Responses to Questions

**Part I - Summary**

Reviewer #1: Human T-cell leukemia virus type 1 (HTLV-1) encodes two viral oncoproteins, Tax and HBZ. In this article, Akkouche et al. show that HBZ inhibits the transforming activities of Tax in vivo using a transgenic Drosophila model. The authors previously reported that the Tax-transgenic Drosophila demonstrated several phenotypes associated with cellular transformation and NF-kappaB activation (Shirinian M. et al. JV, 2015). In the present study, they newly developed HBZ-transgenic Drosophila and a double transgenic strain of Tax and HBZ. Using those animal models, they show that both Tax and HBZ activate PRC2, resulting in the accumulation of H3K27me3 level. Interestingly, HBZ rather suppressed the activation of PRC2 and NF-kappaB, cellular transformation and senescence induced by Tax in their double transgenic flies. Finally, the authors conclude that HBZ and Tax play important roles in the persistence of HTLV-1.

This study was well conducted using the unique animal model. However, several concerns need to be addressed by the authors.

Reviewer #2: The authors demonstrate that antagonistic role of the Human T-Cell Leukemia Virus Type 1 regulatory proteins Tax and HBZ in oncogenic transformation in vivo.

They showed the evidence that antagonistic role of Tax and HBZ in oncogenic transformation in a gene modified fly system. They also performed in vitro experiment to analyze underlying molecular mechanism.

The study contains some new findings about the molecular characterization HTLV-1 viral protein Tax and HBZ in fly system. However, it is little evidence to understand HTLV-1 pathogenesis in human, because there are several missing points between experimental result and evidence observed in ATL patients in previous studies.

Reviewer #3: In the present study, Akkouche et al. investigate the functional role of HTLV-1 HBZ and its interaction with the Tax oncoprotein using their transgenic Drosophila model. The rough eye phenotype and increased hemocyte count are used as indicators of transformation. Results in this model are in part validated using in vitro overexpression experiments in HEK cells and HBZ silencing experiments in the MT-1 cell line.

Results show that, like Tax, HBZ is capable of enhancing activity of the PRC complex, resulting in increased H3K27me3 repressive marks. Unlike Tax, however, HBZ neither induces “transformation”, nor NF-κB activation and instead antagonizes the positive effects of Tax on cellular transformation and NF-kB activation in the Drosophila model.

The paper is clearly written and the experimental setup well laid-out. Although the experiments are well-executed and the results are clear, there is some concern as to how much information can really be gained from the Drosophila model, when many transgenic mouse models for Tax and HBZ, as well as humanized mouse models are available. To this effect, it is puzzling that HBZ alone is oncogenic in mice while it clearly is not in Drosophila.

One must also consider that in > 50% of ATL Tax expression is lost due to 5’ deletions spanning the x-IV ORF or to 5’LTR methylation, while HBZ expression is conserved, suggesting that ATL cells are addicted to HBZ (but not Tax) expression and that, in the context of leukemia cells, HBZ function is likely to be distinct from its capacity to overcome Tax-induced senescence. This suggests that, at least in a significant proportion of leukemia cells, “alleviating the deleterious effects” of the transient bursts of Tax expression is not a likely explanation of HBZ function and is not sufficient to reconcile the apparently contrasting results obtained in the fly model.

In summary, the paper would greatly benefit if the authors would make a stronger point of the advantage of using their Drosophila model, and make an effort to integrate their (partly puzzling) results in a working model.

**Part II – Major Issues: Key Experiments Required for Acceptance**

Reviewer #1: 1. In the transgenic flies, HBZ suppresses PRC2 activity only in the presence of Tax, although it seems that HBZ essentially has a property to activate PRC2. Mechanisms for those contradicting results have not been evaluated enough.

2. The authors suggest that Tax and HBZ competitively bind to EZH2 according to the result shown in Fig 5E. It is possible that Tax and HBZ target the same domain in EZH2 protein. If they can prove it, the result would strengthen their conclusion.

3. As the authors mentioned in the manuscript, it has been reported that HBZ suppresses Tax-induced senescence by inibition of NF-kappaB hyperactivation, and promotes proliferation of Tax-expressing cells (Zhi H. et al. PLOS Pathog, 2011). Therefore, it is easy to understand the significance of NF-kappaB inactivation by HBZ for HTLV-1 persistence. On the other hand, there is almost no description about the meaning of PRC2 inhibition in Tax/HBZ co-expressing cells. How critical is PRC2 suppression by HBZ for Tax-expressing cells?

Reviewer #2: 1. The first question is why they used the fly system. There are some explanations, but they would not be enough to convince readers.

2. It would be better to put appropriate introduction. They describes several previous evidence to understand the roles of Tax and HBZ in ATL leukemogenesis. It is important to know the situation in fresh ATL cells.

Several previous reports demonstrated that Tax expression is frequently silenced but HBZ expression is maintained in ATL cells(doi: 10.1073/pnas.0507631103, doi: 10.1038/ng.3415). Also, there are several evidences about high frequency of defective provirus in 5' side of HTLV-1.

Thus, tax is not transcribed in such ATL cells due to lack of template DNA.

For example, it is misleading to describe "Critically, long-term survival of ATL-derived cells depends on Tax expression [20, 21]. : line 80-81". Because that may be true for some ATL cells, but we cannot generalize that.

3. Regarding role of NF-kB in ATL leukemogenesis.

NF-kB is not the focus of this study, but authors describes about the point to emphasise the role of Tax in ATL lekemogenesis.

There are some logical flaws.

For example. not only viral factors but also host factors can explain activation of NF-kB in ATL cells. Actually a previous study demonstrated that high frequency of activation mutation in TCR signalling pathway (doi: 10.1038/ng.3415). Loss of mir31 expression can activate NF-kB signalling pathway in ATL cells(doi: 10.1016/j.ccr.2011.12.015). It's fair to include these key papers when we discuss about NF-kB and ATL cells. Again, NF-kB is not main topic of this study.

4. One of the key points about HTLV-1-mediated oncogenesis is cell-type specificity. HTLV-1 infects various cell types as well as CD4 T cells, but transform CD4 T cells almost exclusively.

As an advantage of in vivo system, we can analyze effect of cell type specificity by changing promoter to express transgene, tax and HBZ.

To address the points "role of the Human T-Cell Leukemia Virus Type 1 regulatory proteins Tax and HBZ in oncogenic transformation in vivo", I'd like to suggest authors to express viral genes in the CD4 T cells.

Reviewer #3: N/A

**Part III – Minor Issues: Editorial and Data Presentation Modifications**

Reviewer #1: 1. Author summary needs to be revised since the content is quite similar to the Abstract.

2. Page 5, line 109. It’s assumed that a word “dispensable” is wrong. “indispensable”?

3. The authors show that Tax activates both NF-kappaB and PRC2. However, the association between those pathways/molecules is not clear. Yamagishi et al. previously reported that overexpression of PRC2 components suppresses the expression of miR-31 which targets NIK, leading to activation of NF-kappaB pathway in ATL cells (Yamagishi M. et al. Cancer Cell, 2012). Are there any links between NF-kappaB and PRC2 in their Drosophila model? In Fig 2H, it is shown that knockdown of E(z) and Suz12 inhibited the induction of Diptericin induced by Tax; however, it is still obscure what the result means. They should explain about it.

4. In Fig 3D, the number of hemocytes is increased in Tax Tg flies but not in Tax/HBZ Tg. On the other hand, Fig 4D and E show that most of Tax Tg hemocytes (~80%) are senescent, while Tax/HBZ Tg hemocytes seem to be normal. If HBZ cancels cellular senescence induced by Tax, it is expected that hemocytes in the double Tg flies are increased, as Zhi H. et al. reported (Zhi H. et al. PLOS Pathog, 2011). How the authors explain those counterintuitive results?

Reviewer #2: (No Response)

Reviewer #3: In summary, the paper would greatly benefit if the authors would make a stronger point of the advantage of using their Drosophila model, and make an effort to integrate their (partly puzzling) results in a working model.

PLOS authors have the option to publish the peer review history of their article (what does this mean?). If published, this will include your full peer review and any attached files.

Reviewer #1: No

Reviewer #2: No

Reviewer #3: No
---

## [Decision Letter · Decision Letter 1]

23 Nov 2020

Dear Ali,

Thank you very much for submitting your revised manuscript "In vivo antagonistic role of the Human T-Cell Leukemia Virus Type 1 regulatory proteins Tax and HBZ in oncogenic transformation." for consideration at PLOS Pathogens. As with all papers reviewed by the journal, your manuscript was reviewed by members of the editorial board and by several independent reviewers. In light of the reviews (below this email), we would like to invite the resubmission of a significantly-revised version that takes into account the reviewers' comments.

Reviewers 1 and 3 still consider that the results are overinterpreted, because of the doubt over how justifiable it is to extend the results in Drosophila to the manifestly different system in the natural host. Please therefore modify the title and the discussion in line with the comments made by the reviewers, making a clearer distinction between the Drosophila and mammalian systems, and be more circumspect in drawing conclusions as to the pathogenesis in the natural host.

We cannot make any decision about publication until we have seen the revised manuscript and your response to the reviewers' comments. Your revised manuscript is also likely to be sent to reviewers for further evaluation.

With best wishes,

Charles

Charles R. M. Bangham

Associate Editor

PLOS Pathogens

Susan Ross

Section Editor

PLOS Pathogens

Kasturi Haldar

Editor-in-Chief

PLOS Pathogens

orcid.org/0000-0001-5065-158X

Michael Malim

Editor-in-Chief

PLOS Pathogens

orcid.org/0000-0002-7699-2064

Reviewers 1 and 3 consider that the results are overinterpreted, because of the doubt over how justifiable it is to extend the results in Drosophila to the manifestly different system in the natural host. Please therefore modify the title and the discussion in line with the comments made by the reviewers, making a clearer distinction between the Drosophila and mammalian systems, and be more circumspect in drawing conclusions as to the pathogenesis in the natural host.

Reviewer's Responses to Questions

**Part I - Summary**

Reviewer #1: The authors responded to most of the questions.

Reviewer #2: The authors demonstrate that antagonistic role of the Human T-Cell Leukemia Virus Type 1 regulatory proteins Tax and HBZ in oncogenic transformation in vivo.

They showed the evidence that antagonistic role of Tax and HBZ in oncogenic transformation in a gene modified fly system. They also performed in vitro experiment to analyze underlying molecular mechanism.

The finding is really interesting, but there is not enough evidence to support the title "In vivo antagonistic role of the Human T-Cell Leukemia Virus Type 1 regulatory proteins Tax and HBZ in oncogenic transformation".

Reviewer #3: The Authors have significantly improved the quality of their manuscript providing new data to address most of the criticisms raised for their previous submission.

**Part II – Major Issues: Key Experiments Required for Acceptance**

Reviewer #1: None.

Reviewer #2: 1. Justification for the use of the Drosophila model

There are additional experiments using human cells. Tax and/or HBZ is over-expressed in 293T or Jurkat cells. It would be more convincing if the authors would compare the expression level of them with that in vivo in their Drosophila model and primary ATL cells. In the Drosophila model, Tax and HBZ protein are detected by W.B as shown Fig1E. Especially Tax expression level seems to be high. Since we rarely see such protein expression level in vivo in mouse model and infected individuals.

It would be more convincing if the authors show that transgene expression level is close to physiological level in vivo.

2. Viral gene expression in ATL cells.

I do appreciate the authors' effort to analyze PBMCs from 6 ATL cases. However, there is little information about the details of the ATL cases. What is the proportion of ATL cells and nonATL-HTLV-1 infected cells in the PBMCs. It is difficult to know whether the viral gene expression detected comes from ATL cells or non-malignant infected cells. Flowcytometry analysis will be useful to provide such information (doi: 10.1158/1078-0432.CCR-13-3169.).

Reviewer #3: n/a

**Part III – Minor Issues: Editorial and Data Presentation Modifications**

Reviewer #1: Fig 2I: There seems to be faint, but not clear, difference between Tax+/mCherry RNAi (lane 2) and Tax+/Relish RNAi lane 3). Since this result is important to show that PRC2 activity is correlated with NFkB in this animal model, quantification and statistical evaluation of H3K27me3 levels should be done as the authors showed in Fig 1G.

Reviewer #2: (No Response)

Reviewer #3: n/a

PLOS authors have the option to publish the peer review history of their article (what does this mean?). If published, this will include your full peer review and any attached files.

Reviewer #1: No

Reviewer #2: No

Reviewer #3: No
---

## [Editor Report · Decision Letter 2]

4 Dec 2020

Dear Dr. Bazarbachi,

We are pleased to inform you that your manuscript 'In vivo antagonistic role of the Human T-Cell Leukemia Virus Type 1 regulatory proteins Tax and HBZ in oncogenic transformation.' has been provisionally accepted for publication in PLOS Pathogens.

Best regards,

Charles R. M. Bangham

Associate Editor

PLOS Pathogens

Susan Ross

Section Editor

PLOS Pathogens

Kasturi Haldar

Editor-in-Chief

PLOS Pathogens

orcid.org/0000-0001-5065-158X

Michael Malim

Editor-in-Chief

PLOS Pathogens

orcid.org/0000-0002-7699-2064
---

## [Editor Report · Acceptance letter]

5 Jan 2021

Dear Dr. Bazarbachi,

We are delighted to inform you that your manuscript, "In vivo antagonistic role of the Human T-Cell Leukemia Virus Type 1 regulatory proteins Tax and HBZ
," has been formally accepted for publication in PLOS Pathogens.

Best regards,

Kasturi Haldar

Editor-in-Chief

PLOS Pathogens

orcid.org/0000-0001-5065-158X

Michael Malim

Editor-in-Chief

PLOS Pathogens

orcid.org/0000-0002-7699-2064